# PRISM: Festina Lente Proactivity—Risk-Sensitive, Uncertainty-Aware Deliberation for Proactive Agents

**Yuxuan Fu**[1], **Xiaoyu Tan**[2*], **Teqi Hao**[1*], **Chen Zhan**[1], **Xihe Qiu**[1†]
[1]Shanghai University of Engineering Science, [2]National University of Singapore
`yuxuanfu@sues.edu.cn, txywilliam1993@outlook.com, teqihao@gmail.com`
`chenzhan361@gmail.com, xiheqiu1993@gmail.com`

## Abstract

Proactive agents must decide not only *what* to say but also *whether and when* to intervene. Many current systems rely on brittle heuristics or indiscriminate long reasoning, which offers little control over the benefit-burden tradeoff. We formulate the problem as cost-sensitive selective intervention and present PRISM, a novel framework that couples a decision-theoretic gate with a dual-process reasoning architecture. At inference time, the agent intervenes only when a calibrated probability of user acceptance exceeds a threshold derived from asymmetric costs of missed help and false alarms. Inspired by *festina lente* (Latin: "make haste slowly"), we gate by an acceptance-calibrated, cost-derived threshold and invoke a resource-intensive *Slow* mode with counterfactual checks only near the decision boundary, concentrating computation on ambiguous and high-stakes cases. Training uses *gate-aligned, schema-locked distillation*: a teacher running the full PRISM pipeline provides dense, executable supervision on unlabeled interaction traces, while the student learns a response policy that is *explicitly decoupled* from the intervention gate to enable tunable and auditable control. On PROACTIVEBENCH, PRISM reduces false alarms by **22.78%** and improves F1 by **20.14%** over strong baselines. These results show that principled decision-theoretic gating, paired with selective slow reasoning and aligned distillation, yields proactive agents that are precise, computationally efficient, and controllable. To facilitate reproducibility, we release our code, models, and resources at `https://prism-festinalente.github.io/`; all experiments use the open-source PROACTIVEBENCH benchmark.

## 1 Introduction

Proactive agents aim to surface help at the right moment, before users ask, yet without becoming intrusive (Bubeck et al., 2023; Schick et al., 2023; Yao et al., 2023; Achiam et al., 2023; Lu et al., 2024). The central challenge is a speak or remain silent decision under asymmetric costs. False alarms erode trust and add cognitive load, while misses forgo timely assistance. In real deployments this decision must be made with partial information, nontrivial tool latency, and a strict compute budget. Consider a coding copilot. When a flaky continuous integration failure emerges during a sensitive configuration edit, the agent should intervene only if help is needed and likely to be accepted; otherwise it should stay out of the way.

Prevailing pipelines often rely on ad hoc thresholds or chain of thought used by default. This leads to two persistent failures: **over intervention** that interrupts user flow, and **indiscriminate compute** that expends slow and expensive reasoning even for obvious cases (Cobbe et al., 2021; Schick et al., 2023; Wang & Zhou, 2024; Zhang et al., 2024). In addition, acceptance optimization is frequently decoupled from timing control. Prompts or output formats are tuned offline, and only later are heuristics added for when to speak. This blurs the line between the learned policy and product controls and weakens guarantees on the tradeoff between quality and efficiency.

---

*Equal contribution.
†Corresponding author.

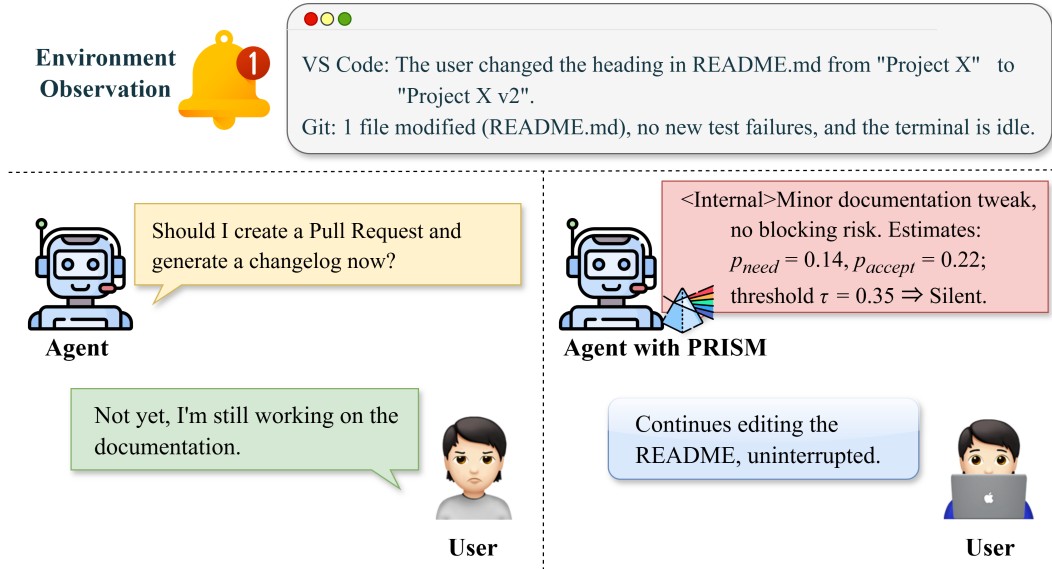

Figure 1: **PRISM reduces false alarms by estimating confidence before speaking.** The agent first estimates calibrated $(\hat{p}_{\text{need}}, \hat{p}_{\text{accept}})$, compares $\hat{p}_{\text{accept}}$ with an adaptive threshold $\tau (\hat{p}_{\text{need}}; C_{\text{FA}}, C_{\text{FN}})$, and triggers a single pass slow stage only within a narrow region near the threshold. In benign edit events, low $\hat{p}_{\text{need}}$ raises $\tau$, the gate remains *silent*, and the user continues without interruption.

We introduce **PRISM** (*Proactive Risk Sensitive Intervention with a Slow mode Margin*), a simple and principled framework that unifies *need and acceptance modeling*, *cost sensitive gating*, and *selective slow reasoning*. This design is inspired by *festina lente* (Latin: "make haste slowly"): the agent gates by expected utility and invokes slow mode only near the decision boundary. We frame timing as a selective decision over two calibrated probabilities: the probability that help is needed ($p_{\text{need}}$) and the probability that help will be accepted ($p_{\text{accept}}$). A cost aware gate compares acceptance confidence with an adaptive threshold that depends on $p_{\text{need}}$ and on the relative costs of false alarms and misses. Slow reasoning is not universal. PRISM allocates a single pass slow stage only near this decision boundary through a slow mode margin, which concentrates computation where it is most likely to change the outcome. The result is an interpretable control surface with a small set of knobs, such as cost terms and the width of the near boundary region, that moves the benefit and burden frontier in predictable ways. This connects to the literature on selective prediction and calibration (El-Yaniv et al., 2010; Geifman & El-Yaniv, 2017; Guo et al., 2017). We further characterize how the adaptive threshold varies monotonically with costs and with $p_{\text{need}}$, and we relate the slow trigger rate to expected latency, which yields a compact map from knobs to metrics.

Training mirrors deployment. PRISM applies the same costs, the same gate, and the same slow mode margin to shape both runtime control and the learning signal, which reduces the gap between simulation and production. Concretely, we mix three terms in a deployment aligned objective: (i) outcome acceptance, (ii) need consistency that rewards silence when no help is needed and penalizes silence when help is needed, and (iii) burden terms that price user disruption and tokens used by slow reasoning. Our evaluation follows paired event protocols that stress timing under partial information and realistic tool use. We report results on synthetic teacher traces with diagnostics and on real world coding logs (Jimenez et al., 2024; Zhang et al., 2024). Code and protocols will be released upon acceptance through an anonymous repository.

**Contributions.** PRISM connects a simple principle with a minimal implementation and a reproducible evaluation protocol:

- **Cost sensitive gating in PRISM.** The method separates need from acceptance and concentrates single pass slow reasoning near an adaptive boundary, which yields a compact and interpretable control surface for quality and efficiency tradeoffs.

- **Gate aligned objective.** The same costs and margins are used at train time and at test time, which improves calibration and closes the gap between simulation and production.

- **Budget aware evaluation.** We report precision, recall, F1, false alarm rate, slow token cost, and area based summaries of the benefit and burden tradeoff (AUDBC). Across always slow, no slow, fixed threshold, and schema only distillation baselines, PRISM reduces false alarms at matched recall while lowering slow compute, and it maintains control stability under distribution shift (Yu et al., 2025). These properties are essential for reliable deployment.

## 2 RELATED WORK

**Proactive agents and event-stream timing.** Recent work shifts from purely reactive assistance to deciding *when* to intervene within evolving event streams. *Proactive Agent*/ProactiveBench formalize acceptance-supervised intervention under realistic workflows with partial information, emphasizing timing and user burden (Lu et al., 2024). Similar challenges in optimizing intervention timing have been observed in critical decision-support domains (Hao et al., 2025), highlighting the universality of this temporal decision problem. We build on this direction and emphasize *deployment alignment*: the gate, costs, and visible schema are shared between training and inference. This design isolates improvements that arise from better timing rather than from interface or prompt mismatch (Lu et al., 2024).

**Risk-sensitive intervention, abstention, and calibration.** The choice to speak or remain silent is a binary decision with asymmetric costs for false alarms and missed help. Cost-sensitive learning and reject-option theory provide principled thresholds and characterize the risk and coverage trade-off (Elkan, 2001; Chow, 2003). Selective deep classifiers instantiate these ideas with integrated abstention heads (Geifman & El-Yaniv, 2019). Because our gate relies on calibrated probabilities, it connects to modern calibration methods for neural networks (Guo et al., 2017) and to metareasoning views that allocate computation where the expected value of computation is high (Russell & Wefald, 1991). Our controller concentrates additional SLOW computation within a near-threshold band, operationalizing these principles for proactive timing.

**Uncertainty-aware objectives beyond binary rewards.** Standard RLHF typically optimizes a single scalar reward. In contrast, we estimate intervention probabilities ($p_{\text{need}}, p_{\text{accept}}$) and compose them with labels ($y_{\text{need}}, y_{\text{accept}}$) into a structured objective over event streams, which is consistent with reinforcement learning principles (Sutton & Barto, 2018). This view aligns with efforts to move beyond binary rewards by training language models to reason about their own uncertainty (Damani et al., 2025). It also dovetails with selective prediction, since these probabilities parameterize a cost-sensitive gate rather than optimizing a monolithic reward (Geifman & El-Yaniv, 2019).Such alignment strategies relate to recent works on using LLM-derived feedback to balance helpfulness and harmlessness (Tan et al., 2023), and frameworks leveraging LLMs to guide reward signals in decision-making tasks (Deng et al., 2025).

**Protocol-aligned synthesis, distillation, and selective SLOW.** To scale supervision while remaining faithful to the evaluation protocol, we synthesize event-conditioned decisions with a teacher language model and distill them into a smaller model (Hinton et al., 2015), following self-generated instruction pipelines (Wang et al., 2023). When uncertainty is high, we invoke a SLOW scratchpad-style pass motivated by chain-of-thought prompting (Wei et al., 2022; Tan et al., 2024) while keeping the visible schema identical across training and evaluation. This alignment ensures that reliability gains stem from improved timing and calibration rather than from format shifts.

## 3 METHOD

We frame proactive assistance as a cost-sensitive selective decision problem. At each timestep $t$, our agent observes a context $X_t$ and estimates two calibrated probabilities: $p_{\text{need},t} = \Pr(\text{help is needed} \mid X_t)$ and $p_{\text{accept},t} = \Pr(\text{offer is accepted} \mid X_t, \text{intervene})$. The decision to intervene is governed by a cost-sensitive gate. Let $C_{\text{FA}}$ be the cost of a false alarm and $C_{\text{FN}}$ be the cost of a false negative. The agent intervenes only if the estimated acceptance probability $p_{\text{accept},t}$ exceeds a dynamic threshold

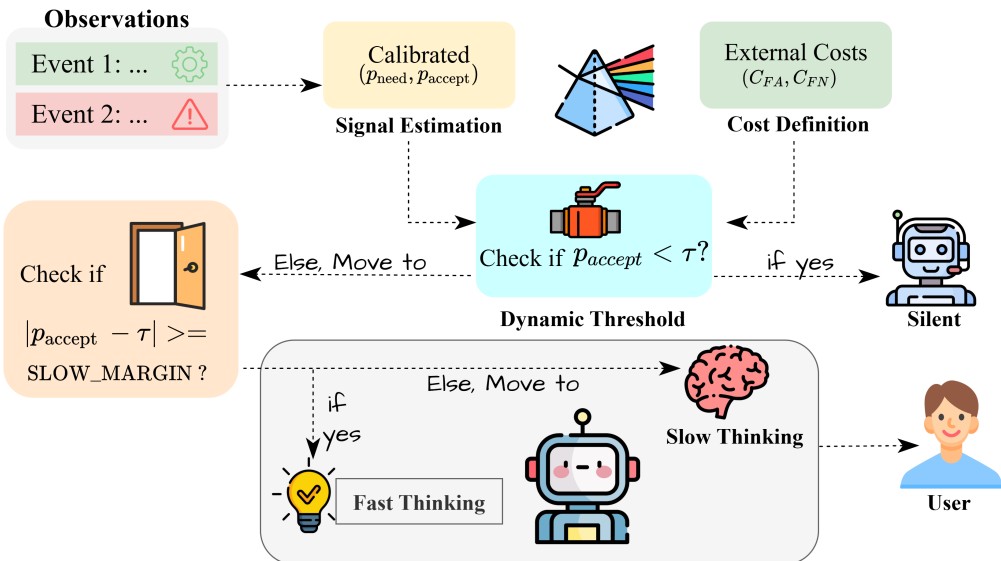

Figure 2: **PRISM Pipeline.** A model first estimates two calibrated probabilities from the context $X_t$: the user's need for assistance ($p_{\text{need}}$) and the likely acceptance of an offer ($p_{\text{accept}}$). A cost-sensitive gate then decides to intervene only if the acceptance probability meets a dynamic threshold $\tau(p_{\text{need}})$ that accounts for the relative costs of false alarms ($C_{\text{FA}}$) and missed opportunities ($C_{\text{FN}}$). To balance accuracy and efficiency, a resource-intensive slow reasoning pass is triggered only when the initial prediction is ambiguous and falls within a narrow margin $\delta_{\text{slow}}$ of the decision boundary.

determined by the estimated need $p_{\text{need},t}$:

$$p_{\text{accept},t} \;\geq\; \tau(p_{\text{need},t}) \;\triangleq\; \frac{C_{\text{FA}}}{C_{\text{FA}} + p_{\text{need},t} \cdot C_{\text{FN}}}. \tag{3.1}$$

This rule ensures that as the certainty of need increases, the agent can tolerate a lower chance of acceptance to intervene. To improve decisions in ambiguous cases without universal high cost, we employ a dual-process architecture. A fast model provides initial estimates $(p_{\text{need},t}^{F}, p_{\text{accept},t}^{F})$. A single, more powerful slow reasoning pass is triggered only if the fast estimate is close to the decision boundary, i.e., $|p_{\text{accept},t}^{F} - \tau(p_{\text{need},t}^{F})| \leq \delta_{\text{slow}}$, where $\delta_{\text{slow}}$ is a configurable margin.

The training process is explicitly aligned with this decision architecture. We employ a method called **Decision-Consistent Curation (RDC)** to distill knowledge from a powerful teacher model. We filter the training data using a ranking score $R_{\text{DC}}$ that rewards the teacher for accepted interventions and penalizes it for miscalibrated probability estimates:

$$R_{\text{DC}} \;=\; y_{\text{accept}} \;-\; \left(q_{\text{need}} - y_{\text{need}}\right)^2 \;-\; \mathbf{1}\{y_{\text{need}}^{(\text{pred})} = 1\}\left(q_{\text{accept}} - y_{\text{accept}}\right)^2, \tag{3.2}$$

where $(q_{\text{need}}, q_{\text{accept}})$ are the teacher's probabilities and $(y_{\text{need}}, y_{\text{accept}})$ are ground-truth labels. A student model is then trained via supervised fine-tuning on the top-ranked, curated data subset $\mathcal{D}^{\star}$. The learning objective $\mathcal{L} = \mathcal{L}_{\text{need}} + \mathcal{L}_{\text{acc}} + \mathcal{L}_{\text{burden}}$ ensures that the student's estimates of $p_{\text{need}}$ and $p_{\text{accept}}$ are well-calibrated (using inverse propensity scoring for $\mathcal{L}_{\text{acc}}$ to handle selection bias) and includes regularization terms that penalize the burden of false alarms and excessive slow reasoning.

## 4 Experiments

We evaluate PRISM on PROACTIVEBENCH, targeting three desiderata of proactive assistance: *timeliness*, *accuracy*, and *cost sensitivity*. Our primary evaluated model is a student trained via *RDC distillation* with supervised fine-tuning (RDC-SFT). We compare against strong open-source and proprietary baselines, and we analyze how PRISM's decision policy shapes the benefit–burden trade-off. Our study addresses three key questions:

**(Q1) Comparative effectiveness & Human Alignment.** How does PRISM perform relative to proprietary frontiers and prior SOTA? Crucially, are the improvements on automatic metrics *validated by human experts*?

**(Q2) Inference Efficiency.** To what extent does the *slow-margin gating* strategy improve the decision quality without incurring the high latency and compute costs of "System 2" reasoning?

**(Q3) Mechanism & Training.** How do *RDC-filtered supervision* and *dual-signal gating* ($p_{\text{need}}, p_{\text{accept}}$) contribute to performance? Is the system robust to calibration drift?

## 4.1 EXPERIMENTAL SETUP

**Training data and recipe.** Our student, QWEN3-8B-PRISM, is trained with full-parameter SFT on an *RDC-filtered* subset of the official training split containing **1,800** instances ($< \frac{1}{3}$ of the original). During training, we use AdamW with a learning rate of $1 \times 10^{-5}$ and a 0.1 warm-up ratio under a cosine schedule for 3 epochs. We adopt the Qwen chat template with a 4096-token context and `bf16` precision; the effective batch size per device is 4 (per-device batch size 1, gradient accumulation 4). Training runs on a single NVIDIA A100 (80 GB) and completes in approximately 2.5 hours. Full hyperparameters are listed in Appendix G.

**Evaluation Dataset.** All experiments use PROACTIVEBENCH (Lu et al., 2024), which contains user event streams from three domains: coding, writing, and daily life. We follow the official split: the training portion is used only for student distillation (and, where applicable, weighted SFT), while the held-out test set (233 clips) is reserved for evaluation. Each clip is a sequence of tuples $(E_t, A_t, S_t)$ for time step $t$, denoting the observed event, user activity, and environment state. At each step, the agent either issues a proactive proposal $P_t$ or remains silent.

**Baselines.** We compare the following systems under a unified evaluation harness.

1. **Frontier API LMs (proprietary).** Strong proprietary models (e.g., GPT–4o, Claude 3.5–Sonnet) prompted for proactive assistance without any specialized gating or distillation.

2. **Open-source base LMs.** Strong open models (e.g., LLaMA–3.1–8B, Qwen2–7B) with the same prompting protocol and no special gating/distillation.

3. **Proactive Agent (SOTA).** The public pipeline and decision policy from the original PROACTIVEBENCH paper, evaluated as provided.

4. **PRISM.** A student trained via *RDC* distillation with supervised fine-tuning (RDC-SFT), evaluated with the full PRISM decision policy.

**Metrics.** We report standard acceptance-aware detection metrics on the test streams: *Recall*, *Precision*, *Accuracy*, *False-Alarm rate*, and *F1* with a numerical stabilizer $\varepsilon > 0$:

$$\text{F1} = \frac{2 \cdot \text{Precision} \cdot \text{Recall}}{\text{Precision} + \text{Recall} + \varepsilon}.$$

To summarize benefit–burden trade-offs we adopt a risk–coverage style view from selective prediction and instantiate it with a cost-sensitive utility, yielding AUDBC as our area-under-curve summary (Geifman & El-Yaniv, 2017).We do not train a reward model. Instead, we adopt an LLM-as-Judge setup that strictly follows the evaluation guidelines. Let $C_{\text{FA}} > 0$ be the per-false-alarm cost and $C_{\text{FN}} \geq 0$ the per-missed-opportunity (false negative) cost. For a fixed $C_{\text{FN}}$, a decision policy induces (i) a normalized burden $\mathcal{B}(C_{\text{FN}}) \in [0, 1]$ (we use the false-alarm rate) and (ii) a normalized net *benefit* $\Delta\mathcal{U}(C_{\text{FN}}) \in [0, 1]$ defined as the utility improvement over the *always-silent* baseline after accounting for $C_{\text{FA}}$ and $C_{\text{FN}}$. We summarize performance by integrating $\Delta\mathcal{U}$ along the burden axis:

$$\text{AUDBC} = \int_0^1 \Delta\mathcal{U}(b)\, \mathrm{d}b \ \in [0, 1], \tag{4.1}$$

$$\text{with } \Delta\mathcal{U}(b) = \frac{\mathbb{E}[\text{TP}] - C_{\text{FA}}\mathbb{E}[\text{FP}] - C_{\text{FN}}\mathbb{E}[\text{FN}] - \mathbb{E}[\text{TP}]_{\text{silent}}}{Z}, \quad b \equiv \mathcal{B}(C_{\text{FN}}), \tag{4.2}$$

| Model | Recall ↑ | Precision ↑ | Accuracy ↑ | False-Alarm ↓ | F1-Score ↑ |
|---|---|---|---|---|---|
| *Automatic Evaluation (LLM-as-Judge)* | | | | | |
| Claude-3.5-Sonnet | 97.89% | 45.37% | 49.78% | 54.63% | 62.00% |
| GPT-4o-mini | **100.00%** | 35.28% | 36.12% | 64.73% | 52.15% |
| GPT-4o | 98.11% | 48.15% | 49.78% | 51.85% | 64.60% |
| LLaMA-3.1-8B-Proactive | 99.06% | 49.76% | 52.86% | 50.24% | 66.25% |
| Qwen2-7B-Proactive | **100.00%** | 49.78% | 50.66% | 50.22% | 66.47% |
| Deepseek-R1 | 98.12% | 72.35% | 72.96% | 27.64% | 83.28% |
| Qwen3-8B | 73.79% | 73.33% | 67.85% | 26.67% | 73.34% |
| **Qwen3-8B-PRISM** | 98.88% | **77.05%** | **76.39%** | **22.94%** | **86.61%** |
| *Human Expert Evaluation* | | | | | |
| DeepSeek-R1 | 99.05% | 70.03% | 71.12% | 29.70% | 82.05% |
| **Qwen3-8B-PRISM** | **99.41%** | **74.01%** | **73.79%** | **25.91%** | **84.85%** |

Table 1: **Main comparison on PROACTIVEBENCH (held-out test set).** We report performance under two protocols: *Automatic Evaluation* (using our validated LLM-ensemble) and *Human Expert Evaluation* (bottom rows). PRISM (our RDC-distilled student with dynamic gating) consistently outperforms strong proprietary and open-source baselines across both regimes. Notably, it surpasses its teacher (*DeepSeek-R1*) in Precision and False-Alarm rate while maintaining near-perfect Recall, validated by human experts.

where TP, FP, FN are expectations per time step under the evaluated policy, $\mathbb{E}[\text{TP}]_{\text{silent}}$ is the always-silent baseline utility (zero for detection tasks), and $Z > 0$ normalizes to $[0, 1]$.[1] A higher AUDBC indicates consistently larger net benefit at the same or lower burden across cost settings.

**Judging and Validation Protocol.** We do not train a reward model. Instead, we adopt an LLM-as-Judge setup that *strictly* follows the official PROACTIVEBENCH rubric. At each step $t$, we employ a majority vote among three strong frontier models—*DeepSeek-R1*, *GPT-4o*, and *Claude 3.5-Sonnet*—to determine whether help is *needed* and likely to be *accepted*. The gold label is derived as $y^{\star}(t) = y_{\text{need}}(t) \wedge y_{\text{accept}}(t)$. Acknowledging recent findings on potential biases and overconfidence in LLM judges (Ye et al., 2024), we rigorously validated our protocol against human judgment. We sampled 229 events and enlisted 7 volunteers from diverse backgrounds (including undergraduate and graduate students in programming, marketing, and business) to annotate intervention needs. Our ensemble judge achieved 89.1% agreement with human consensus and a Cohen's $\kappa$ of 0.71 (substantial agreement), confirming that it serves as a reliable proxy for human evaluation in this domain. Detailed agreement statistics and comparisons across different judge pools are provided in Appendix C.

## 4.2 MAIN RESULTS: PROACTIVE ASSISTANCE PERFORMANCE

Table 1 demonstrates that PRISM establishes a new state-of-the-art by effectively curbing the "over-proactiveness" plague inherent in existing models.

**Solving the False-Alarm Dilemma.** Baseline agents, including strong proprietary models (e.g., GPT-4o) and prior SOTA (Qwen2-7B-Proactive), suffer from excessive chatter, with false-alarm rates consistently exceeding 50%. PRISM drastically corrects this behavior. Compared to Qwen2-7B-Proactive, it improves the *F1-Score* by over 20 points ($66.47 \rightarrow 86.61$) while cutting the *False-Alarm Rate* by nearly 54% relative ($50.22 \rightarrow 22.94$). This massive gain in precision ($49.78 \rightarrow 77.05$) is achieved with negligible impact on the near-saturated recall.

**Statistically Significant Superiority.** Crucially, the PRISM-trained student does not merely imitate its teacher; it surpasses it. Despite using a smaller backbone, our model improves upon the powerful *DeepSeek-R1* teacher. Statistical significance testing (details in Appendix D) confirms that while

---

[1]In practice we sweep $C_{\text{FN}}$ over a grid, obtain pairs $\left(\mathcal{B}(C_{\text{FN}}^{(i)}), \Delta\mathcal{U}(C_{\text{FN}}^{(i)})\right)$ sorted by burden, and estimate equation 4.1 via the trapezoidal rule: $\text{AUDBC} \approx \sum_i \frac{1}{2}\left(\Delta\mathcal{U}_{i+1} + \Delta\mathcal{U}_i\right)\left(b_{i+1} - b_i\right)$.

| Policy | Recall ↑ | Precision ↑ | Accuracy ↑ | False-Alarm ↓ | F1-Score ↑ |
|---|---|---|---|---|---|
| Fixed $\tau{=}0.5$ | **92.42%** | **71.96%** | **76.03%** | **28.23%** | **80.74%** |
| Static-cost $\tau(C_{\mathrm{FA}}, C_{\mathrm{FN}})$ | 38.54% | 59.67% | 63.94% | 40.32% | 46.83% |
| Dynamic $\tau(p_{\mathrm{need}})$ | 72.41% | 68.29% | 69.52% | 31.70% | 70.29% |

Table 2: **Ablation A: thresholding only (no gating).** Fixed threshold dominates when $p_{\mathrm{need}}$ is uncalibrated.

PRISM maintains **parity in Recall** ($+0.76\%$, $p = 0.115$), it achieves **statistically significant improvements** in *Precision* ($+4.70\%$, $p < 0.001$) and *False-Alarm Rate* reduction ($p < 0.001$). This confirms that the precision gains are not random fluctuations but a result of our structured distillation.

**Human-Verified Reliability.** This advantage is robust to the evaluation protocol. As shown in the bottom section of Table 1, human experts corroborate the automatic metrics: PRISM delivers a superior user experience, achieving a higher F1-score than DeepSeek-R1 ($84.85\%$ vs. $82.05\%$) and a lower false-alarm rate ($25.91\%$ vs. $29.70\%$). This alignment between LLM and human judges (Cohen's $\kappa = 0.71$) underscores the validity of our results.

**Attribution.** We attribute these gains to two factors: (1) **Need/Accept Disentangling**: RDC-SFT teaches the student to explicitly estimate $p_{\mathrm{accept}}$, filtering out "correct but unwanted" proposals that the teacher might generate. (2) **Cost-Sensitive Gating**: The dynamic threshold and slow-margin gate route only borderline cases to deliberation, recovering true positives without the computational cost or hallucination risk of universal reasoning (Deng et al., 2026).

## 4.3 Ablation Studies

We categorize our analyses into four parts using the Qwen3-8B backbone: (A) evaluates thresholding on the *base* model to demonstrate the need for calibration; (B) analyzes the efficiency-quality trade-off of our inference strategy; (C) compares training paradigms; and (D) dissects the contribution of decision signals and robustness to cost/parameter shifts.

**(A) Thresholds (policy).** We compare three decision rules under **direct inference** (no RDC-SFT, no post-hoc calibration): (a) *Fixed* threshold, (b) *Static cost* threshold $\tau(C_{\mathrm{FA}}, C_{\mathrm{FN}})$, (c) *Dynamic* threshold $\tau(p_{\mathrm{need}})$ computed from the model's (uncalibrated) $p_{\mathrm{need}}$.

*Analysis.* The *fixed* rule yields the best overall balance (F1 80.74) and lowest false alarms (28.23). The *static-cost* rule is overly conservative on this distribution, collapsing recall (38.54) and harming F1. The *dynamic* $\tau(p_{\mathrm{need}})$ improves markedly over static-cost (F1 70.29 vs. 46.83) but still underperforms fixed, with higher false alarms (31.70 vs. 28.23).

This pattern is consistent with **probability miscalibration**: without RDC-SFT or post-hoc calibration, $p_{\mathrm{need}}$ is noisy near the decision boundary, so instance-wise thresholding does not reliably separate needed from non-needed steps. In our main results, where $p_{\mathrm{need}}$ (and $p_{\mathrm{accept}}$) are calibrated via RDC-SFT, dynamic policies surpass fixed thresholds. Here, the ablation isolates that *better policies require calibrated signals*; Post-hoc temperature scaling is a simple but surprisingly effective fix for probability miscalibration, and we observe similar gains here when calibrating on a held-out judge-labeled set (Guo et al., 2017).

**(B) Inference Strategy (Efficiency).** We examine whether PRISM can deliver "System 2" quality at "System 1" speeds. Table 3 compares strategies using the RDC-SFT student.

*Analysis.* Table 3 reveals a compelling efficiency-quality trade-off, positioning PRISM as a Pareto improvement over static inference strategies. By dynamically routing only $\sim$11% of borderline cases to the slow reasoning path, PRISM achieves a massive quality boost—improving F1 by **+5.06 points** over *Fast-only* ($83.09\% \rightarrow 88.15\%$)—while incurring a negligible P95 latency overhead of just **20ms**. We selected the margin $\delta = 0.1$ based on a detailed sensitivity analysis (see Appendix E and Figure 5), which identifies it as the "sweet spot" before latency costs scale linearly. As visualized in Figure 3, our method defines the optimal Pareto frontier (blue line). This frontier is derived from a comprehensive 2D grid sweep across varying cost ratios and margins (fully detailed in Appendix

| Strategy | Recall ↑ | Precision ↑ | False-Alarm ↓ | F1-Score ↑ | AUDBC ↑ | Tokens | Lat (P95) |
|---|---|---|---|---|---|---|---|
| Fast-only | **100%** | 71.08% | 28.92% | 83.09% | 79.26% | **510** | **176ms** |
| Slow-only | **100%** | 75.17% | 24.83% | 86.79% | 81.43% | 693 | 312ms |
| **Slow-on-margin** | **100%** | **78.81%** | **21.19%** | **88.15%** | **82.72%** | 541 | 196ms |

Table 3: **Ablation B: Efficiency.** Slow-on-margin ($\delta = 0.1$) achieves the best performance with negligible latency overhead compared to Fast-only.

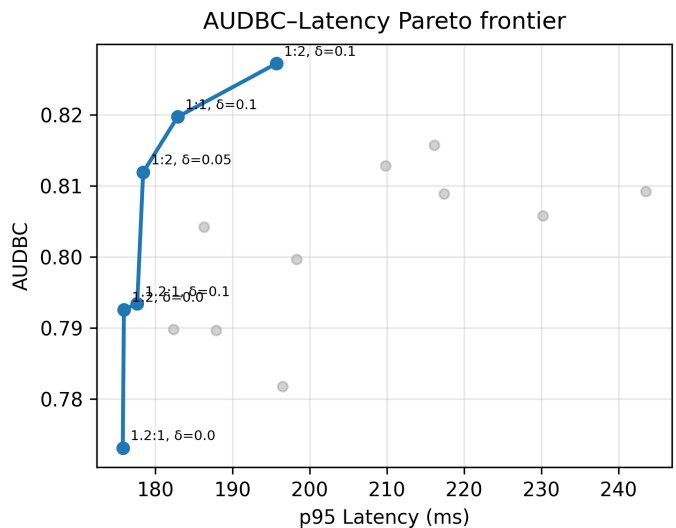

Figure 3: **Efficiency-Quality Pareto Frontier.** We plot AUDBC against P95 Latency across various cost ratios ($C_{\text{FA}} : C_{\text{FN}}$) and slow-margins ($\delta$). The **blue line** traces the optimal frontier, showing that PRISM (particularly at $\delta = 0.1$) achieves maximal decision benefit with minimal latency overhead ($\sim$20ms). Grey points represent suboptimal configurations (e.g., higher $\delta$ or uncalibrated baselines) that incur higher latency without corresponding utility gains.

Table 10), demonstrating that margin-based gating effectively captures the utility gains of "System 2" reasoning without the prohibitive latency of *Slow-only* execution.

**(C) Distillation and SFT pathways (training).** We compare four training pathways on the **same backbone (Qwen3-8B)** with matched decoding and judging. (i) **SFT**: vanilla supervised fine-tuning on the *unfiltered* dataset (no RDC screening). (ii) **DFT**: *Dynamic Fine-Tuning* Wu et al. (2025), which stabilizes token-level gradients by dynamically rescaling the objective with each token's model probability; in our setting DFT is applied as a *second-stage* fine-tune starting from the RDC-SFT checkpoint. (iii) **Weighted-SFT**: a second-stage fine-tune from RDC-SFT using an RL-like scalar weight $R_{\text{total}}$ for each example, adapted from Reward-Weighted Regression (RWR) Peters & Schaal (2007); Peng et al. (2019). (iv) **RDC-SFT (ours)**: full-parameter SFT on an *RDC-filtered* dataset with explicit ($p_{\text{need}}, p_{\text{accept}}$) supervision.

*Analysis.* RDC-SFT attains the best overall balance: it raises *F1* to 86.61 (vs. 76.09 for SFT, $+10.52$ pts; vs. 80.59 for Weighted-SFT, $+6.02$ pts) and yields the lowest *False-Alarm* (22.49, a reduction of $-5.07$ vs. SFT and $-6.82$ vs. Weighted-SFT), while keeping *Recall* high (98.88). Vanilla SFT on the unfiltered corpus achieves perfect recall but at the cost of higher chatter (FA 27.56) and lower precision (72.43), reflecting label noise and timing mismatches. Weighted-SFT improves *F1* over SFT, but its scalar weights $R_{\text{total}}$—which depend on uncalibrated ($q_{\text{need}}, q_{\text{accept}}$)—inflate false alarms (FA 29.31), indicating sensitivity to noise and acceptance miscalibration. Second-stage DFT underperforms here (Recall 80.14): when applied *after* RDC-SFT, dynamic rescaling can blunt acceptance-aware gradients learned during distillation, yielding conservative triggering without precision gains (same 72.43 as SFT).

| Training Pathway | Recall ↑ | Precision ↑ | Accuracy ↑ | False-Alarm ↓ | F1-Score ↑ |
|---|---|---|---|---|---|
| SFT | **100.00%** | 72.43% | 69.52% | 27.56% | 76.09% |
| DFT | 80.14% | 72.43% | 69.52% | 27.56% | 76.09% |
| Weighted-SFT | 93.70% | 70.68% | 72.10% | 29.31% | 80.59% |
| **RDC-SFT(ours)** | 98.88% | **77.05%** | **76.39%** | **22.49%** | **86.61%** |

Table 4: **Ablation C: training paradigms on Qwen3-8B.** RDC-SFT (ours) trains on RDC-filtered supervision with explicit need/accept targets; DFT and Weighted-SFT are *second-stage* finetunes starting from the RDC-SFT checkpoint.

Overall, these results suggest that **data quality and target structure dominate**: RDC filtering plus explicit $(p_{\text{need}}, p_{\text{accept}})$ supervision improves precision and suppresses false alarms without materially sacrificing recall, whereas post-hoc reweighting (Weighted-SFT) or probability-rescaled objectives (DFT) are less effective in the presence of acceptance/timing noise.

**(D) Decision Mechanism & Calibration.** We isolate the contribution of our dual gating signals $(p_{\text{need}}, p_{\text{accept}})$ and quantify the impact of post-hoc calibration on the RDC-SFT student.

| Configuration | Recall ↑ | Precision ↑ | False-Alarm ↓ | F1-Score ↑ | AUDBC ↑ |
|---|---|---|---|---|---|
| $p_{\text{accept}}$ only ($p_{\text{need}}=1$) | 99.95% | 46.20% | 62.50% | 63.19% | 58.40% |
| $p_{\text{need}}$ only ($p_{\text{acc}}=\tau(p_{\text{need}})$) | 99.15% | 69.50% | 29.10% | 81.72% | 82.45% |
| Uncalibrated $(p_{\text{need}}, p_{\text{accept}})$ | 98.80% | 74.77% | 25.23% | 85.12% | 86.43% |
| **Calibrated (Ours)** | **98.88%** | **77.05%** | **22.94%** | **86.61%** | **88.52%** |

Table 5: **Ablation D: Gating Signals & Calibration.** Using both signals is crucial for reducing false alarms. Post-hoc calibration further aligns probabilities with risk, boosting Precision and AUDBC.

*Analysis.* Table 5 validates two key components of PRISM. First, **Need/Accept Disentanglement**: Relying solely on $p_{\text{accept}}$ (Row 1) causes a catastrophic rise in False Alarms (62.50%) because users often accept helpful but ill-timed suggestions. Using $p_{\text{need}}$ alone (Row 2) is safer but suboptimal. Combining them (Rows 3-4) balances timeliness with acceptance. Second, **Calibration Utility**: While the uncalibrated RDC-SFT model is already strong (Row 3), post-hoc temperature scaling (Row 4) significantly reduces the False-Alarm rate ($25.23\% \rightarrow 22.94\%$) and improves AUDBC ($+2.09$). This precise probabilistic alignment also underpins the model's robustness to varying cost ratios ($C_{\text{FA}} : C_{\text{FN}}$) and domain shifts, as detailed in Appendix F.

## 4.4 CASE STUDY: STRUCTURED TRACES THAT MOTIVATE PRISM

**Observation.** We first benchmark a reasoning-capable model (DeepSeek-R1) on ProactiveBench and observe substantially stronger proactive assistance than non-reasoning baselines. To probe what drives this gap *without exposing free-form scratchpads*, we elicit *structured* traces via the PROACTIVE AGENT workflow: We analyze only these structured fields offline under de-identified, non-user-facing settings.

**Decomposable decision pattern.** Across representative traces, we find that high-quality assistance co-occurs with two separable signals: (i) whether the user likely *needs* intervention at the current event, and (ii) conditional on proposing, whether the suggestion is likely to be *accepted*. We formalize these signals as probabilities $p_{\text{need}} \in [0, 1]$ and $p_{\text{accept}} \in [0, 1]$, estimated from the structured outputs above (e.g., `Purpose`/`Proactive Task`/`Response`) via a small calibrator trained on held-out judgments.[2]

**Design implication.** This decomposition motivates PRISM: a lightweight gate that triggers intervention when $p_{\text{need}}$ is confident, and selectively invokes slow-mode deliberation near decision margins; meanwhile the final offer is regulated by the acceptance propensity $p_{\text{accept}}$, enabling explicit control of burden–benefit trade-offs. Figure 4 illustrates a typical trace and its mapping to $(p_{\text{need}}, p_{\text{accept}})$ used by our gate.

---

[2]No free-form chain-of-thought is stored or surfaced; analysis uses only structured fields and aggregate labels.

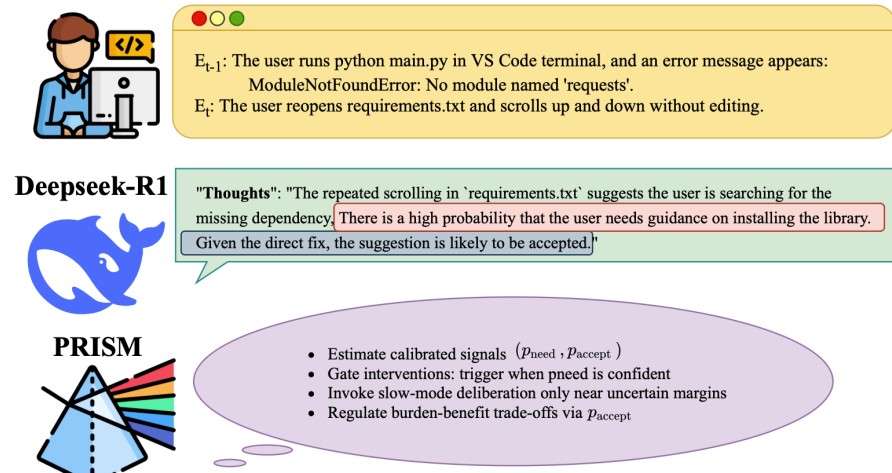

Figure 4: **Structured trace motivating PRISM.** A ProactiveBench coding event (top) yields a *structured* output from a reasoning-capable teacher (DeepSeek-R1; middle). PRISM (bottom) abstracts the teacher's implicit hints into calibrated signals $p_{\text{need}}$ and $p_{\text{accept}}$, uses a lightweight gate to trigger interventions when $p_{\text{need}}$ is confident, and invokes slow-mode deliberation only near uncertain margins; the final offer is regulated by $p_{\text{accept}}$. Only structured fields are analyzed; no free-form chain-of-thought is stored or surfaced.

## 5   DISCUSSION

**Reframing Proactivity as Calibrated Control.** Current proactive systems often conflate the *capability* to generate a solution with the *decision* to intervene, leading to the "over-eagerness" observed in baselines. PRISM demonstrates that proactivity is better modeled as a dual-variable control problem: disentangling the probability of need ($p_{\text{need}}$) from acceptance ($p_{\text{accept}}$). This separation allows the agent to remain silent even when it can technically solve a problem, simply because the social or cognitive cost of interruption outweighs the probabilistic benefit.

**Economic Rationality in Neural Reasoning.** Our *festina lente* approach operationalizes the "System 1 vs. System 2" paradigm through an economic lens. Rather than treating long-chain reasoning as a default mode, PRISM treats it as a scarce resource to be allocated only when the expected utility of reducing uncertainty exceeds the computational cost. The success of our slow-on-margin gating (11% slow rate achieving Pareto optimality) suggests that for proactive agents, *uncertainty estimation is as critical as reasoning capability*.

**Data Efficiency and Control Surfaces.** Finally, our results challenge the trend of indiscriminately scaling supervision. By filtering for calibrated probabilities via RDC, we show that a compact student can surpass a stronger teacher in alignment-critical metrics such as precision and false-alarm rate. Crucially, PRISM exposes an interpretable control surface—defined by cost terms ($C_{\text{FA}}, C_{\text{FN}}$) and margin $\delta$—that bridges static training with dynamic deployment, allowing developers to tune agent assertiveness for different user personas without retraining.

## 6   LIMITATIONS AND FUTURE WORK

Despite rigorous validation on PROACTIVEBENCH, reliance on LLM-as-Judge proxies may diverge from real-world user acceptance and overlook the cognitive load of ill-timed interruptions during flow states. Methodologically, PRISM assumes offline probability calibration remains stable; however, severe deployment distribution shifts could degrade $p_{\text{need}}$ and $p_{\text{accept}}$ accuracy, making dynamic thresholds suboptimal without online recalibration. Furthermore, our formulation assumes stationary costs ($C_{\text{FA}}, C_{\text{FN}}$) during interactions. Realistically, missed intervention costs ($C_{\text{FN}}$) rise near deadlines, while false alarm penalties ($C_{\text{FA}}$) drop as users become visibly stuck. Implementing time-varying costs remains challenging online since total episode lengths are unknown. Future work could address this via a meta-controller that adapts cost sensitivity based on progress or implicit feedback. Finally, distillation processes may inherit teacher biases, requiring hard safety constraints so utility-based gating does not suppress critical security warnings. Future iterations could also integrate PRISM with advanced agent workflows and tool-use optimization frameworks (Tan et al., 2025) to handle complex, multi-step maintenance tasks.

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

# APPENDIX

## ETHICS STATEMENT

This work does not present any significant ethical concerns. All datasets used in our research, such as ProactiveBench, are publicly available and well-established academic benchmarks. The data are anonymized and does not contain personally identifiable information. Our proposed method, which focuses on improving model efficiency and accuracy, does not have foreseeable direct negative societal impacts. No human subjects were involved in our experiments.

## STATEMENT ON LLM USAGE

During the preparation of this work, we used LLMs, such as ChatGPT, for assistance with language editing, grammar correction, and improving readability. All core ideas, methodologies, experimental designs, results, and conclusions were originally conceived and articulated by the human authors. We have carefully reviewed and edited all text generated or modified by LLMs and take full responsibility for the final content of this paper. LLMs were not listed as authors.

# A   DERIVING THE COST-SENSITIVE THRESHOLD

We provide a self-contained derivation of the gate in Eq. (3.1) and its odds form Eq. (A.2) from Bayes-risk minimization under asymmetric costs.

**Proposition 1** (Bayes-optimal gate under asymmetric costs)**.** *Let $p_{need} = \Pr(Y_{need}{=}1 \mid X_t)$ and $p_{accept} = \Pr(Y_{accept}{=}1 \mid X_t)$. Assume costs $C_{FA} > 0$ for proposing when $Y_{accept} = 0$ (false alarm) and $C_{FN} > 0$ for staying silent when help is needed and would be accepted (missed help), with correct decisions costing zero. Then the Bayes-optimal decision is*

$$\text{INTERVENE} \iff p_{accept} \geq \tau\big(p_{need}\big), \quad \tau(p_{need}) = \frac{C_{FA}}{C_{FA} + p_{need}\, C_{FN}}. \tag{A.1}$$

*Equivalently (odds form),*

$$\frac{p_{accept}}{1 - p_{accept}} \geq \frac{p_{need}\, C_{FN}}{C_{FA}}. \tag{A.2}$$

*Proof.* Condition on $X_t$ and write expected costs for $a \in \{\text{int}, \text{sil}\}$. Intervening pays only on a false alarm:

$$\mathbb{E}[\text{Cost} \mid a = \text{int}, X_t] = (1 - p_{\text{accept}})\, C_{FA}. \tag{A.3}$$

Staying silent pays only on a missed help; the effective FN term scales with $p_{\text{need}}$:

$$\mathbb{E}[\text{Cost} \mid a = \text{sil}, X_t] = p_{\text{accept}}\,(p_{\text{need}}\, C_{FN}). \tag{A.4}$$

Choose the action with smaller conditional expectation. Intervene iff

$$(1 - p_{\text{accept}})C_{FA} \leq p_{\text{accept}}\, p_{\text{need}}\, C_{FN},$$

which rearranges to Eq. (A.2) and hence to Eq. (A.1).  □

**Comparative statics.**   From $\tau(p_{\text{need}}) = \dfrac{C_{FA}}{C_{FA} + p_{\text{need}}C_{FN}}$,

$$\frac{\partial \tau}{\partial p_{\text{need}}} = -\frac{C_{FA} C_{FN}}{(C_{FA} + p_{\text{need}}C_{FN})^2} < 0, \quad \frac{\partial \tau}{\partial C_{FA}} > 0, \quad \frac{\partial \tau}{\partial C_{FN}} < 0.$$

Thus higher burden $C_{FA}$ tightens the gate, while higher risk $C_{FN}$ or higher need $p_{\text{need}}$ relax it.

# B   AUDBC (AREA UNDER $\Delta$-BURDEN CURVE)

For each evaluation event $e$, let $q_e \in [0, 1]$ denote the model's estimated need probability $p_{\text{need}}$, $p_e \in [0, 1]$ the estimated acceptance probability $q_{\text{accept}}$, and $n_e \in \mathbb{N}$ the number of candidate proposals. Given a fixed false-alarm cost $C_{FA} > 0$ and a false-negative cost $C_{FN} > 0$, the *dynamic threshold* used by our gate is

$$\tau(q_e; C_{FA}, C_{FN}) = \frac{C_{FN} \cdot q_e}{C_{FA} + C_{FN} \cdot q_e},$$

which is the default `"odds"` implementation in our code.[3] An intervention is triggered for event $e$ under cost $C_{FN}$ iff

$$\mathbb{I}_{C_{FN}}(e) = \mathbf{1}[p_e \geq \tau(q_e; C_{FA}, C_{FN}) \,\wedge\, n_e > 0].$$

For a set $\mathcal{E}$ of $N$ evaluable events, the *normalized burden* and *expected benefit* at $C_{FN}$ are

$$B(C_{FN}) = \frac{1}{N} \sum_{e \in \mathcal{E}} \mathbb{I}_{C_{FN}}(e), \qquad U(C_{FN}) = \frac{1}{N} \sum_{e \in \mathcal{E}} \mathbb{I}_{C_{FN}}(e) \cdot p_e.$$

We sweep $C_{FN}$ over a predefined grid $\{c_k\}_{k=1}^K$ (environment variable `AUDBC_CFN_GRID`) with $C_{FA}$ fixed (env `COST_FA`), obtaining points $\big(B(c_k), U(c_k)\big)$. After removing duplicates and sorting by $B$, we compute AUDBC by the trapezoidal rule on $[0, 1]^2$:

$$\text{AUDBC} \approx \sum_{k=1}^{K-1} \big(B_{k+1} - B_k\big) \cdot \frac{U_k + U_{k+1}}{2} \quad \in [0, 1],$$

where $B_k = B(c_k)$ and $U_k = U(c_k)$. Higher AUDBC indicates better benefit–burden trade-off under cost-sensitive gating.

---

[3]The implementation flag `AUDBC_TAU_IMPL` defaults to `"odds"`; we also support a `"bayes"` variant, but all reported results use `"odds"`.

## C    Evaluation Protocol and Judge Validation Details

To ensure the reliability of our automated evaluation, we conducted a rigorous validation study comparing our LLM-as-Judge protocol against both human experts and alternative LLM panels.

### C.1    Judge Pools and Human Annotation

We enlisted 7 volunteers from diverse backgrounds (including undergraduate and graduate students in programming, marketing, and business) to annotate a sampled subset of 229 events. The human "Gold" label was determined via majority vote.

For automated evaluation, we validated our primary judge (the majority vote used in the main paper) and further defined two distinct LLM pools to test for robustness:

- **Pool A (Frontier):** GPT-4o, Claude 3.5-Sonnet, and Gemini 2.5-Flash.
- **Pool B (Open):** DeepSeek-R1, Kimi-K2, and Llama-3-70B.

### C.2    Agreement Analysis

We measured the alignment between our automated judges and the human consensus using Agreement Rate, Cohen's $\kappa$, and Matthews Correlation Coefficient (MCC).

As shown in Table 6, our primary ensemble judge achieves substantial agreement with human labels ($\kappa = 0.71$). Furthermore, the high consistency across different pools (Pool A vs. Pool B $\kappa = 0.76$) confirms that the evaluation signal is stable and not dependent on idiosyncratic biases of specific model families.

| Comparison Setting | Support | Agreement ↑ | Cohen's $\kappa$ ↑ | MCC ↑ |
|---|---|---|---|---|
| *Main Protocol Verification* | | | | |
| Our Ensemble Judge vs. Human | 229 | 89.1% | 0.71 | 0.71 |
| *Cross-Model Consistency Checks* | | | | |
| Pool A (Frontier) vs. Human | 229 | 90.8% | 0.75 | 0.76 |
| Pool B (Open) vs. Human | 229 | 89.1% | 0.71 | 0.71 |
| Pool A vs. Pool B | 229 | 91.2% | 0.76 | 0.77 |

Table 6: **Comprehensive Human-Model and Inter-Model Agreement Analysis.** "Our Ensemble Judge" refers to the majority vote protocol (GPT-4o, Claude-3.5-Sonnet, DeepSeek-R1) used for the main results. The high agreement scores ($> 0.70\kappa$) across all settings confirm the reliability of the automated evaluation protocol.

### C.3    Robustness of Performance Gains

A critical concern in LLM-based evaluation is whether performance improvements are an artifact of the specific judge employed. To address this, we evaluated the relative performance of PRISM against the strongest baseline, DeepSeek-R1, across all three evaluation regimes.

Table 7 summarizes these results. While absolute F1 scores fluctuate—consistent with observations by Ye et al. (2024) regarding varying judge severity—the **relative improvement ($\Delta$F1)** remains highly robust. PRISM consistently outperforms the baseline by approximately **+3.0 F1 points** across all evaluators, including human judges.

## D    Statistical Significance Analysis

To verify that the performance gains of PRISM over its teacher model (DEEPSEEK-R1) are not due to random chance, we conducted statistical significance testing using bootstrap resampling ($N = 10,000$ iterations). Table 8 reports the 95% confidence intervals for the performance deltas and the corresponding $p$-values.

The analysis confirms that PRISM's primary contribution—suppressing false alarms to improve precision—is highly statistically significant ($p < 0.001$), demonstrating robust behavior distinct from the teacher model's base capabilities.

| Judge Pool | Method | F1-Score ↑ | ΔF1 (Gain) ↑ |
|---|---|---|---|
| Pool A (Frontier) | DeepSeek-R1 | 82.92% | – |
| | **PRISM (Ours)** | **86.05%** | **+3.13%** |
| Pool B (Open) | DeepSeek-R1 | 83.28% | – |
| | **PRISM (Ours)** | **86.61%** | **+3.33%** |
| Human (Majority) | DeepSeek-R1 | 82.05% | – |
| | **PRISM (Ours)** | **84.85%** | **+2.80%** |

Table 7: **Performance comparison across diverse judges.** PRISM maintains a consistent advantage over the DeepSeek-R1 baseline regardless of whether the evaluator is a proprietary model ensemble, an open-source ensemble, or human experts.

| Metric | DeepSeek-R1 | PRISM (Ours) | Imp. | 95% CI | P-value |
|---|---|---|---|---|---|
| **Recall ↑** | 98.12% | 98.88% | +0.76% | $[-0.18\%, +1.75\%]$ | 0.115 |
| **Precision ↑** | 72.35% | 77.05% | +4.70% | $[+2.35\%, +7.05\%]$ | $< 0.001$ |
| **Accuracy ↑** | 72.69% | 76.39% | +3.43% | $[+1.10\%, +5.76\%]$ | 0.004 |
| **False-Alarm ↓** | 27.64% | 22.94% | -4.70% | $[-7.12\%, -2.28\%]$ | $< 0.001$ |
| **F1-Score ↑** | 83.28% | 86.61% | +3.33% | $[+1.52\%, +5.15\%]$ | 0.002 |

Table 8: **Statistical significance of PRISM vs. DeepSeek-R1.** The results confirm that the improvements in Precision, Accuracy, False-Alarm rate, and F1 are statistically significant ($p < 0.05$), while Recall remains statistically equivalent.

# E    DETAILED ANALYSIS OF INFERENCE STRATEGY

In this section, we provide the detailed numerical results supporting the inference strategy ablation discussed in §4.3. We analyze the sensitivity of the slow-margin hyperparameter $\delta$ and the robustness of the system across varying cost ratios ($C_{\text{FA}} : C_{\text{FN}}$).

## E.1    IMPACT OF SLOW MARGIN

We investigated the impact of the confidence margin threshold $\delta$ on the efficiency-accuracy trade-off. The results are detailed in Table 9 and visualized in Figure 5.

Table 9: **Sensitivity sweep for slow-margin $\delta$.** A margin of $\delta = 0.1$ strikes the optimal balance, achieving the highest F1 and AUDBC. Increasing $\delta$ further to $0.15$ yields diminishing returns in accuracy while significantly increasing latency and token consumption.

| Margin $\delta$ | Rec ↑ | Pre ↑ | Acc ↑ | FA ↓ | F1 ↑ | AUDBC ↑ | Tokens | Lat |
|---|---|---|---|---|---|---|---|---|
| 0 (Fast) | 100% | 71.08% | 71.07% | 28.92% | 83.09% | 79.26% | 510 | 176ms |
| 0.05 | 100% | 74.11% | 74.10% | 25.89% | 85.12% | 81.19% | 518 | 178ms |
| **0.10** | **100%** | **78.81%** | **79.33%** | **21.19%** | **88.15%** | **82.72%** | 541 | 196ms |
| 0.15 | 100% | 74.57% | 75.20% | 25.43% | 85.43% | 80.58% | 563 | 230ms |

## E.2    ROBUSTNESS ACROSS COST RATIOS

To verify that our method is not over-fitted to a specific cost setting, we conducted a 2D grid sweep over the cost ratio $C_{\text{FA}} : C_{\text{FN}}$ (ranging from eager 1:4 to conservative 1.2:1) and the slow margin $\delta$. Table 10 presents the full results.

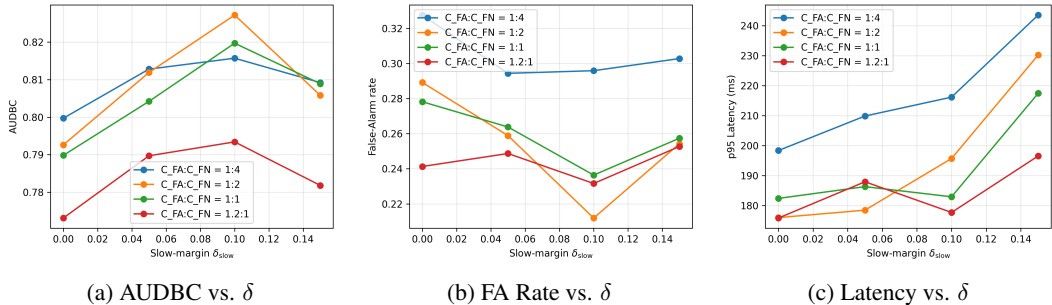

(a) AUDBC vs. $\delta$            (b) FA Rate vs. $\delta$           (c) Latency vs. $\delta$

Figure 5: **Impact of Slow Margin.** Increasing the margin initially boosts performance by correcting hard examples, but latency grows linearly. $\delta = 0.1$ is the "sweet spot".

Table 10: **Robustness Grid Sweep.** Performance metrics across varying user cost profiles ($C_{\text{FA}}$ : $C_{\text{FN}}$) and slow margins. PRISM consistently benefits from the margin-based gating ($\delta > 0$) across diverse cost settings, maintaining high F1 and AUDBC scores.

| Cost Ratio ($C_{\text{FA}} : C_{\text{FN}}$) | Margin ($\delta$) | Rec ↑ | Pre ↑ | Acc ↑ | FA ↓ | F1 ↑ | AUDBC | Tokens | Lat |
|---|---|---|---|---|---|---|---|---|---|
| 1 : 4 | 0 | 100% | 67.24% | 78.93% | 32.76% | 83.64% | 79.97% | 532 | 198ms |
| | 0.05 | 100% | 70.56% | 81.24% | 29.44% | 82.13% | 81.28% | 547 | 210ms |
| | 0.1 | 100% | 70.41% | 80.22% | 29.59% | 83.63% | 81.57% | 562 | 216ms |
| | 0.15 | 100% | 69.72% | 81.57% | 30.28% | 82.96% | 80.92% | 590 | 244ms |
| 1 : 2 | 0 | 100% | 71.08% | 71.07% | 28.92% | 83.09% | 79.26% | 510 | 176ms |
| | 0.05 | 100% | 74.11% | 74.10% | 25.89% | 85.12% | 81.19% | 518 | 178ms |
| | 0.1 | 100% | 78.81% | 79.33% | 21.19% | 88.15% | 82.72% | 541 | 196ms |
| | 0.15 | 100% | 74.57% | 75.20% | 25.43% | 85.43% | 80.58% | 563 | 230ms |
| 1 : 1 | 0 | 98.89% | 72.18% | 80.34% | 27.82% | 83.34% | 78.98% | 517 | 182ms |
| | 0.05 | 97.96% | 73.62% | 81.50% | 26.38% | 84.07% | 80.42% | 532 | 186ms |
| | 0.1 | 98.88% | 72.36% | 80.35% | 23.64% | 83.96% | 81.97% | 534 | 183ms |
| | 0.15 | 98.85% | 74.26% | 78.64% | 25.74% | 83.42% | 80.89% | 568 | 217ms |
| 1.2 : 1 | 0 | 86.18% | 75.87% | 79.90% | 24.13% | 82.39% | 77.31% | 504 | 176ms |
| | 0.05 | 90.34% | 75.13% | 78.44% | 24.87% | 81.28% | 78.97% | 515 | 188ms |
| | 0.1 | 89.62% | 76.83% | 79.59% | 23.16% | 83.74% | 79.34% | 528 | 178ms |
| | 0.15 | 86.45% | 74.73% | 78.52% | 25.27% | 83.45% | 78.18% | 572 | 197ms |

# F ROBUSTNESS AND CALIBRATION ANALYSIS

In this section, we provide detailed supporting evidence for the robustness of PRISM across three dimensions: cost sensitivity, domain generalization, and calibration stability.

## F.1 SENSITIVITY TO COST RATIOS ($C_{\text{FA}} : C_{\text{FN}}$)

Different deployment scenarios require different trade-offs between helpfulness (Recall) and intrusiveness (False Alarm). We swept the cost ratio $C_{\text{FA}} : C_{\text{FN}}$ from $1 : 4$ to $1.2 : 1$ to evaluate how the system adapts to varying penalties for interruptions versus missed opportunities.

As shown in Table 11, PRISM demonstrates strong adaptability without requiring retraining:

1. **Performance Trade-off:** As the penalty for false alarms increases (moving from $1 : 4$ to $1.2 : 1$), the model effectively shifts its operating point. Precision improves ($70.41\% \rightarrow 76.83\%$) and the False-Alarm rate drops significantly ($29.59\% \rightarrow 23.17\%$), while F1 scores remain stable ($> 83\%$) across all settings.

2. **Efficiency Gains:** Stricter constraints ($1.2 : 1$) also lead to greater efficiency. The "Slow Rate" (percentage of calls routed to heavy reasoning) decreases from $15.45\%$ to $9.96\%$, resulting in reduced token consumption and lower latency ($216\text{ms} \rightarrow 177\text{ms}$).

| $C_{FA} : C_{FN}$ | Rec ↑ | Pre ↑ | Acc ↑ | FA ↓ | F1 ↑ | AUDBC ↑ | Slow | Tokens | Lat |
|---|---|---|---|---|---|---|---|---|---|
| 1 : 4 | 100% | 70.41% | 80.22% | 29.59% | 83.63% | 81.57% | 15.45% | 562.43 | 216.16 |
| 1 : 2 | 100% | 73.07% | 81.36% | 26.93% | 84.44% | 82.22% | 13.47% | 538.73 | 191.82 |
| 1 : 1 | 98.88% | 72.36% | 80.35% | 23.64% | 83.96% | 81.97% | 11.15% | 533.79 | 182.90 |
| 1.2 : 1 | 89.62% | 76.83% | 79.59% | 23.17% | 83.74% | 79.34% | 9.96% | 528.34 | 177.67 |

Table 11: **Impact of Cost Ratios.** PRISM adapts to stricter penalties for false alarms (1.2 : 1) by naturally increasing precision and reducing chatter, maintaining high F1 scores across the spectrum.

## F.2 DOMAIN GENERALIZATION

To evaluate whether PRISM learns generalized proactive semantics rather than memorizing domain-specific patterns (e.g., coding syntax), we tested the model on the held-out "Writing" domain (Out-of-Domain), while the training data was dominated by "Coding" tasks (In-Domain).

Table 12 compares the performance of the raw RDC-SFT model against the full PRISM pipeline (with post-hoc Temperature Scaling, denoted as "After-T").

| Domain | Stage | Recall ↑ | Precision ↑ | False-Alarm ↓ | F1-Score ↑ | AUDBC ↑ |
|---|---|---|---|---|---|---|
| In-Domain | Raw | 98.80% | 74.77% | 25.23% | 85.12% | 86.43% |
| (Coding) | **After-T** | 98.80% | 74.77% | 25.23% | 85.12% | 86.43% |
| Out-of-Domain | Raw | 95.83% | 71.39% | 28.61% | 81.37% | 82.15% |
| (Writing) | **After-T** | 98.37% | 73.37% | 26.63% | 82.63% | 84.66% |

Table 12: **Domain Generalization & Calibration Effect.** While the raw model shows a performance dip on the unseen Writing domain, post-hoc calibration ("After-T") effectively recovers Recall (+2.54%) and AUDBC (+2.51%), narrowing the gap between in-domain and out-of-domain performance.

**Analysis.** The results demonstrate strong generalization. Even without domain-specific training, the student model maintains high F1 (> 82%) on writing tasks, suggesting the learned representations for $p_{need}$ and $p_{accept}$ are semantically robust. The raw model tends to be slightly under-confident or miscalibrated on out-of-distribution data (lower Recall of 95.83%). Temperature scaling plays a crucial role here: it corrects the probability distribution, recovering Recall to **98.37%** and significantly boosting AUDBC to **84.66%**. This confirms that our proposed calibration mechanism acts as a stabilizer against domain shifts.

## F.3 CALIBRATION STABILITY AND QUALITY

We provide a two-fold analysis of our post-hoc calibration strategy: first, quantifying the improvement in probability estimation (ECE and Brier score), and second, stress-testing the decision policy against hyperparameter drift.

**Calibration Quality.** Table 13 compares the calibration metrics before and after applying temperature scaling. We optimized separate temperatures for $p_{need}$ ($T_{need} = 0.5$) and $p_{accept}$ ($T_{accept} = 0.7$) on the validation set. The results show a significant reduction in Expected Calibration Error (ECE) for both signals (21.53% → 12.16% for *need*, 13.19% → 7.42% for *accept*). This confirms that the performance gains in Ablation D stem from more reliable probability estimates, ensuring the model is not merely "lucky" at a specific threshold but is genuinely functionally calibrated.

**Sensitivity to Hyperparameter Drift.** To ensure our method is not brittle, we evaluated performance under suboptimal calibration parameters. We perturbed the temperature $T$ (scaling logits) and the decision bias $\epsilon$ (shifting thresholds) around the optimal grid. Table 14 demonstrates strong robustness:

1. **Moderate Drift (Robust):** Within a reasonable range ($T \in [0.75, 1.25]$, $\epsilon \in [-0.15, +0.15]$), F1 scores remain consistently high (> 82.8%). The "Flip Rate" (percentage of decisions changed relative to baseline) remains moderate (< 15%), indicating the core policy is stable.

2. **Extreme Scenarios (Failure Modes):** Only under "super optimistic" ($T = 0.5, \epsilon = +0.3$) or "super pessimistic" ($T = 1.5, \epsilon = -0.3$) settings does performance degrade significantly. This confirms that

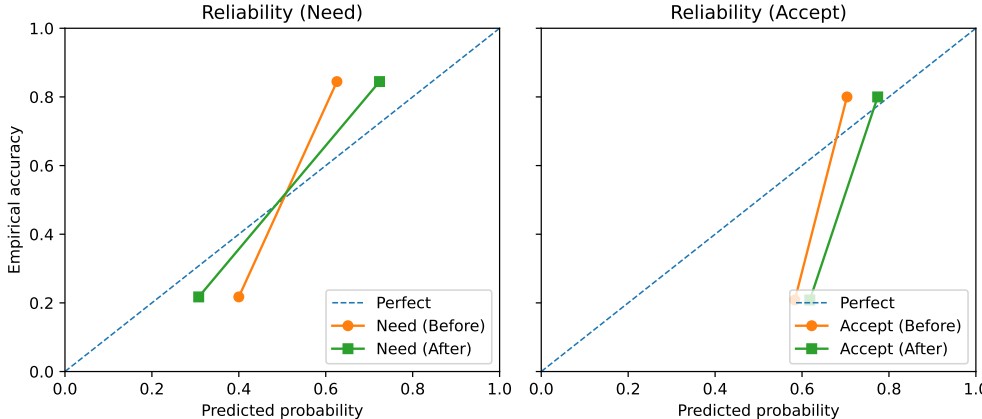

Figure 6: **Reliability Diagrams.** Visualization of calibration performance for Need (left) and Accept (right) probabilities. The green curves (After Scaling) align significantly closer to the perfect diagonal than the orange curves (Before Scaling), visually confirming the ECE improvements.

| | Temperature | | ECE ↓ | | Brier ↓ | | Task Metrics | |
|---|---|---|---|---|---|---|---|---|
| **Setting** | $T_{\text{need}}$ | $T_{\text{acc}}$ | Need | Acc | Need | Acc | F1-Score ↑ | False Alarm ↓ |
| Before Scaling | 1.0 | 1.0 | 21.53% | 13.19% | 17.95% | 18.58% | 84.85% | 25.99% |
| After Scaling | 0.5 | 0.7 | 12.16% | 7.42% | 14.71% | 18.20% | 85.70% | 25.12% |

Table 13: **Calibration Quality Metrics.** Post-hoc scaling significantly reduces Expected Calibration Error (ECE) and Brier scores for both gating signals, creating a more reliable foundation for decision-making.

while calibration helps, the underlying RDC-SFT model provides a stable enough signal that precise tuning is beneficial but not strictly mandatory for viability.

| Scenario | $T$ | $\epsilon$ | Recall ↑ | Pre ↑ | FA ↓ | F1 ↑ | AUDBC ↑ | Flip |
|---|---|---|---|---|---|---|---|---|
| Baseline | 1.0 | 0 | 98.80% | 74.77% | 25.23% | 85.12% | 86.43% | 0% |
| *Moderate Drift (Robustness Region)* | | | | | | | | |
| Moderate | 0.75 | 0 | 98.95% | 74.41% | 25.89% | 84.78% | 84.47% | 6.25% |
| Moderate | 1.25 | 0 | 97.89% | 75.58% | 24.42% | 85.56% | 87.02% | 4.46% |
| Moderate | 1.0 | +0.15 | 99.98% | 71.87% | 28.13% | 83.34% | 81.57% | 12.08% |
| Moderate | 1.0 | -0.15 | 96.67% | 76.22% | 23.78% | 85.30% | 87.34% | 14.83% |
| Moderate | 0.75 | +0.15 | 99.14% | 73.69% | 26.31% | 82.84% | 80.58% | 14.29% |
| Moderate | 0.75 | -0.15 | 97.63% | 75.21% | 24.79% | 85.67% | 85.83% | 13.52% |
| Moderate | 1.25 | +0.15 | 98.73% | 73.33% | 26.67% | 83.21% | 83.24% | 15.17% |
| Moderate | 1.25 | -0.15 | 95.58% | 77.67% | 22.33% | 82.97% | 80.32% | 17.85% |
| *Extreme Scenarios (Failure Modes)* | | | | | | | | |
| Super Optimistic | 0.5 | +0.30 | 100% | 53.18% | 46.82% | 68.67% | 42.57% | 68.34% |
| Super Pessimistic | 1.5 | -0.30 | 37.49% | 88.74% | 11.26% | 38.89% | 45.64% | 72.95% |

Table 14: **Sensitivity Analysis under Parameter Drift.** The model maintains high F1-Score ($> 82\%$) across a wide grid of calibration parameters. Flip denotes the percentage of decisions changed relative to the baseline decision set.

## G  TRAINING CONFIGURATION

```
1  # --- Base model ---
2  model_name_or_path: Qwen3-8B
3  trust_remote_code: true
4  template: qwen
5  cutoff_len: 4096
6  dataset: proactive_agent_rdc_distilled
7  max_samples: 100000
8  train_on_prompt: false
9  per_device_train_batch_size: 1
10 gradient_accumulation_steps: 4
11 learning_rate: 1.0e-5
12 num_train_epochs: 3
13 lr_scheduler_type: cosine
14 warmup_ratio: 0.1
15 pure_bf16: true
```

