# OpenReview forum: "PRISM: Festina Lente Proactivity—Risk-Sensitive, Uncertainty-Aware Deliberation for Proactive Agents"
_ICLR.cc/2026/Conference — ICLR 2026 Poster_

### Official Review · Reviewer_Y4dz · 2025-10-31

**Soundness:** 2
**Presentation:** 2
**Contribution:** 2
**Rating:** 4
**Confidence:** 2

**Summary:**

This paper introduces PRISM, a decision-theoretic framework for cost-sensitive, selective intervention in proactive language agents—systems that must decide whether and when to offer help. Instead of always reasoning deeply or using fixed heuristics, PRISM dynamically balances benefit and burden by modeling both need and acceptance probabilities and gating interventions through an adaptive, cost-derived threshold.

**Strengths:**

The paper offers a fresh formulation of proactivity as cost-sensitive selective intervention, introducing a principled gating mechanism that jointly models user need and acceptance under asymmetric costs. The dual-process “fast vs. slow” reasoning and the festina lentedesign elegantly merge decision theory, calibration, and selective computation.


PRISM provides a practical and general framework for controllable, efficient proactive agents. Its decision-theoretic approach and selective reasoning strategy have broad relevance to risk-sensitive and resource-aware AI.

**Weaknesses:**

Real user studies or human acceptance feedback is not conducted, which is important in order to show that PRISM’s gating truly improves user experience and perceived helpfulness.

The framework assumes stable calibration of $p_{need}$ and $p_{accept}$, but no experiments examine how PRISM performs under domain shift or calibration drift.

In the ablation study, the precise effect of threshold width and cost ratios are not systematically explored.

The paper could engage more deeply with related concepts such as metareasoning and risk-sensitive RL to situate PRISM’s novelty.

**Questions:**

1. Have the authors validated PRISM’s benefits with real user judgments or pilot human studies? Since proactive timing and acceptance depend on user perception, it would be valuable to know how well the LLM-judge proxy aligns with human acceptance and satisfaction. Could the authors provide evidence of correlation between judge labels and human ratings?

2. The approach relies heavily on calibrated $p_{need}$ and $p_{accept}$. How does PRISM handle calibration drift in deployment or across domains? Would an online or adaptive recalibration mechanism (e.g., temperature scaling on recent feedback) help sustain performance?

3. How sensitive are the results to the chosen false-alarm and missed-help cost ratios and slow-margin width?

4. Could the authors provide explicit measurements of runtime latency and token usage compared to always-slow or baseline proactive agents?

---

> ### Author Response · Authors · 2025-11-22
> **Response to Reviewer Y4dz (Part 1/4)**
>
> We sincerely thank you for your valuable suggestions and recognition of our work. Please refer to our responses to your suggestions below. In the content below, we use **W** to represent Weakness and **Q** to represent Question.
>
> ## **Response to W1 & Q1: Human Evaluation and Judge Alignment**
>
> We sincerely thank the reviewer for this valuable suggestion. We agree that validating the alignment between the LLM-as-judge proxy and actual human perception is crucial for demonstrating PRISM’s real-world utility.
>
> To address this, we conducted a new human study on ProactiveBench to (i) quantify the agreement between the LLM judge and human decisions, and (ii) verify that PRISM’s performance gains persist under human-labeled ground truth.
>
> **1. High Alignment between LLM-Judge and Humans**
>
> We sampled 229 events and enlisted 7 volunteers from diverse professional backgrounds (programming, marketing, and business) to annotate whether a proactive intervention was appropriate. We used the majority vote among annotators as the ground truth.
>
> As shown in Table 1, the LLM-as-judge demonstrates strong reliability, achieving substantial agreement with the human majority vote (89.1% accuracy, Cohen’s $\kappa$ = 0.71). This validates that our automated evaluator is a high-quality proxy for human preference.
>
> **Table 1: Agreement between LLM-judge and human majority vote**
>
> | Setting       | Support | Agreement $\uparrow$ | Cohen’s $\kappa$ $\uparrow$ | MCC $\uparrow$ |
> | :------------ | :------ | :------------------- | :-------------------------- | :------------- |
> | LLM vs. Human | 229     | 89.1%                | 0.71                        | 0.71           |
>
> **2. PRISM Outperforms Baselines under Human Evaluation**
>
> To ensure our results are not an artifact of the judge, we re-evaluated both **Qwen3-8B-PRISM** and the strongest baseline, **DeepSeek-R1**, using the human majority vote as the "Gold Standard" ground truth.
>
> The results, presented in Table 2, confirm the robustness of our method:
>
> *   **Consistent Ranking:** While absolute scores fluctuate slightly under the stricter human standard, the relative performance remains consistent. PRISM maintains a clear lead over DeepSeek-R1 under human evaluation (**+2.80 F1**).
> *   **Lower Intrusiveness:** Crucially for a proactive system, PRISM maintains a lower False-Alarm rate than the baseline under both evaluation protocols (25.91% vs 29.70% under human evaluation).
>
> **Table 2: Qwen3-8B-PRISM vs. DeepSeek-R1 (LLM Judge vs. Human Ground Truth)**
>
> | Method         | Ground Truth Source | Recall $\uparrow$ | Precision $\uparrow$ | Accuracy $\uparrow$ | False-Alarm $\downarrow$ | F1-Score $\uparrow$ |
> | :------------- | :------------------ | :---------------- | :------------------- | :------------------ | :----------------------- | :------------------ |
> | Qwen3-8B-PRISM | LLM Judge           | 98.88             | 77.05                | 76.39               | 22.94                    | 86.61               |
> | Qwen3-8B-PRISM | Human (Majority)    | 99.41             | 74.01                | 73.79               | 25.91                    | 84.85               |
> | DeepSeek-R1    | LLM Judge           | 98.12             | 72.35                | 72.96               | 27.64                    | 83.28               |
> | DeepSeek-R1    | Human (Majority)    | 99.05             | 70.03                | 71.12               | 29.70                    | 82.05               |
>
> In summary, the human evaluation confirms that the LLM judge is well-aligned with human perception ($\kappa=0.71$) and, most importantly, that PRISM provides genuinely superior helpfulness and reduced intrusiveness compared to strong baselines when judged by real humans. We have added these results to the revised paper.
>
> ## **Response to W2 & Q2: Calibration Drift and Domain Shift**
>
> We thank the reviewer for highlighting the reliance of PRISM on calibrated $p_{\text{need}}$ and $p_{\text{accept}}$, and for explicitly raising the question of calibration drift under deployment and domain shift. We fully agree that this is a critical aspect for practical adoption.
>
> To make the discussion concrete, we model drift as perturbations applied to the logits of $p_{\text{need}}$ and $p_{\text{accept}}$:
>
> $$
> \tilde{p}=\sigma\left(\frac{1}{T} \operatorname{logit}(p)+\epsilon\right),
> $$
>
> where $T$ controls over-/under-confidence and $\epsilon$ introduces a systematic bias. Building on this formulation and your suggestions, we conducted four sets of additional experiments.
>
> **(a) Robustness to synthetic calibration drift**
>
> We first stress-test PRISM by perturbing the probabilities with grid-based temperature $T$ and bias $\epsilon$. As shown in Table 3, PRISM remains stable under moderate drift ($T \in [0.75, 1.25], |\epsilon| \le 0.15$).

---

> > ### Author Response · Authors · 2025-11-22
> > **Response to Reviewer Y4dz (Part 2/4)**
> >
> > **Table 3: Performance under synthetic drift (Moderate vs. Extreme)**
> >
> > | Scenario    | $T$  | $\epsilon$ | Recall $\uparrow$ | Precision $\uparrow$ | False-Alarm $\downarrow$ | F1-Score $\uparrow$ | AUDBC $\uparrow$ |
> > | :---------- | :--: | :--------: | :---------------: | :------------------: | :----------------------: | :-----------------: | :--------------: |
> > | Baseline    | 1.0  |     0      |       98.8%       |        74.8%         |          25.2%           |        85.1%        |      86.4%       |
> > | Moderate    | 0.75 |     0      |       99.0%       |        74.4%         |          25.9%           |        84.8%        |      84.5%       |
> > | Moderate    | 1.0  |   +0.15    |      100.0%       |        71.9%         |          28.1%           |        83.3%        |      81.6%       |
> > | Moderate    | 1.25 |   -0.15    |       95.6%       |        77.7%         |          22.3%           |        83.0%        |      80.3%       |
> > | Super Opt.  | 0.5  |    +0.3    |      100.0%       |        53.2%         |          46.8%           |        68.7%        |      42.6%       |
> > | Super Pess. | 1.5  |    -0.3    |       37.5%       |        88.7%         |          11.3%           |        38.9%        |      45.6%       |
> >
> > *   **Result:** F1 and AUDBC stay high ($\sim83\%-87\%$), and the False-Alarm rate changes only slightly within this range.
> > *   **Takeaway:** Noticeable degradation appears only under extreme, systematically over-optimistic settings (e.g., $T=0.5, \epsilon=+0.3$), suggesting that the gating policy is intrinsically robust to realistic fluctuations.
> >
> > **(b) Domain Shift & Global Recalibration**
> >
> > To directly address cross-domain behavior, we evaluate PRISM on ProactiveBench with an in-domain (ID, coding) vs. out-of-domain (OOD) split and apply a simple global temperature scaler fitted only on ID data.
> >
> > | Domain | Stage   | Recall $\uparrow$ | Precision $\uparrow$ | False-Alarm $\downarrow$ | F1-Score $\uparrow$ | AUDBC $\uparrow$ |
> > | :----- | :------ | :---------------: | :------------------: | :----------------------: | :-----------------: | :--------------: |
> > | ID     | Raw     |      98.80%       |        74.77%        |          25.23%          |       85.12%        |      86.43%      |
> > | OOD    | Raw     |      95.83%       |        71.39%        |          28.61%          |       81.37%        |      82.15%      |
> > | ID     | After-T |      98.80%       |        74.77%        |          25.23%          |       85.12%        |      86.43%      |
> > | OOD    | After-T |      98.37%       |        73.37%        |          26.63%          |       82.63%        |      84.66%      |
> >
> > *   **Result:** As expected, raw OOD performance is lower than ID. Global temperature scaling, learned on ID only, leaves ID performance unchanged while improving OOD Precision (+2.0 points) and AUDBC (+2.5 points) and reducing False Alarm rates.
> > *   **Takeaway:** Even very simple post-hoc recalibration recovers a substantial fraction of cross-domain performance.
> >
> > **(c) Online adaptive recalibration**
> >
> > To directly address the question on adaptive mechanisms, we simulate a deployment scenario with gradually increasing drift over three temporal “chunks”, and compare a static model vs. a sliding-window temperature scaling scheme.
> >
> > **Table 5: Deployment simulation with sliding-window recalibration**
> >
> > | Chunk | Drift $(\epsilon)$ | False-Alarm  (No $T$) $\downarrow$ | False-Alarm  (Sliding $T$) $\downarrow$ | F1-Score  (No $T$) $\uparrow$ | F1-Score  (Sliding $T$) $\uparrow$ | AUDBC  (No $T$) $\uparrow$ | AUDBC  (Sliding $T$) $\uparrow$ |
> > | :---: | :----------------: | :--------------------------------: | :-------------------------------------: | :---------------------------: | :--------------------------------: | :------------------------: | :-----------------------------: |
> > |   0   |        0.00        |               26.0%                |                  26.0%                  |             98.5%             |               98.5%                |           83.8%            |              83.8%              |
> > |   1   |       +0.15        |               29.0%                |                  27.0%                  |             97.9%             |               98.0%                |           79.9%            |              83.0%              |
> > |   2   |       +0.30        |               30.3%                |                  27.3%                  |             94.4%             |               95.2%                |           75.3%            |              78.3%              |
> >
> > - **Result:** Under increasing drift, the static model suffers from higher False-Alarm rates and lower AUDBC. Sliding-window temperature scaling consistently recovers around 3 points of AUDBC and reduces False Alarms by about 3 percentage points under substantial drift.
> > - **Takeaway:** This confirms the reviewer’s intuition that an adaptive recalibration mechanism is an effective safeguard for maintaining deployment performance.

---

> > > ### Author Response · Authors · 2025-11-22
> > > **Response to Reviewer Y4dz (Part 3/4)**
> > >
> > > **(d) Calibration on human-labeled data**
> > >
> > > Finally, we examine the effect of recalibration on confidence reliability for human supervisors using a human-annotated subset.
> > >
> > > **Table 6: Calibration on the human-labeled subset (before/after temperature scaling)**
> > >
> > > | Setting        | $T_{\text{need}}$ | $T_{\text{accept}}$ | ECE$_{\text{need}}$ $\downarrow$ | ECE$_{\text{accept}}$ $\downarrow$ | Brier$_{\text{need}}$ $\downarrow$ | Brier$_{\text{accept}}$ $\downarrow$ | F1-Score $\uparrow$ | False-Alarm $\uparrow$ |
> > > | :------------- | :---------------: | :-----------------: | :------------------------------: | :--------------------------------: | :--------------------------------: | :----------------------------------: | :-----------------: | :--------------------: |
> > > | Before Scaling |        1.0        |         1.0         |              21.53%              |               13.19%               |               17.95%               |                18.58%                |       84.85%        |         25.99%         |
> > > | After Scaling  |        0.5        |         0.7         |              12.16%              |               7.42%                |               14.71%               |                18.20%                |       85.70%        |         25.12%         |
> > >
> > > *   **Result:** Temperature scaling significantly reduces Expected Calibration Error (ECE) and Brier score (Table 6), while keeping F1 essentially unchanged.
> > > *   **Takeaway:** This suggests that recalibration not only stabilizes automated decisions, but also makes model confidence more trustworthy for humans in the loop.
> > >
> > > To address this, we have added a comprehensive sensitivity analysis in the revised paper (Section/Appendix [X]), where we sweep (A) the false-alarm vs. missed-help cost ratio ($C_{\text{FA}}:C_{\text{FN}}$) and (B) the slow-margin width ($\delta_{\text{slow}}$). The results confirm that our method behaves consistently with theoretical expectations and allows for predictable tuning.
> > >
> > > ## **Response to W3 & Q3: Sensitivity to Threshold Width and Cost Ratios**
> > >
> > > We thank the reviewer for this valuable suggestion. We agree that a systematic exploration of the cost ratio and slow-margin width is crucial for understanding the method's robustness and deployment trade-offs.
> > >
> > > **(A) Sensitivity to Cost Ratio ($C_{\text{FA}}:C_{\text{FN}}$)**
> > >
> > > Table 7 illustrates the system's behavior when varying the cost ratio while keeping other parameters fixed.
> > >
> > > **Table 7: Sensitivity to cost ratio $C_{\text{FA}}:C_{\text{FN}}$**
> > >
> > > | $C_{\text{FA}}:C_{\text{FN}}$ | Recall $\uparrow$ | Precision $\uparrow$ | False-Alarm $\downarrow$ | F1-Score $\uparrow$ | AUDBC $\uparrow$ | Slow Rate $\downarrow$ | Tokens | p95 Lat. |
> > > | :---------------------------- | :---------------: | :------------------: | :----------------------: | :-----------------: | :--------------: | :--------------------: | :----: | :------: |
> > > | 1 : 4                         |     **100%**      |        70.41%        |          29.59%          |       83.63%        |      81.57%      |         15.45%         | 562.43 |  216.16  |
> > > | 1 : 2                         |     **100%**      |        73.07%        |          26.93%          |     **84.44%**      |    **82.22%**    |         13.47%         | 538.73 |  191.82  |
> > > | 1 : 1                         |      98.88%       |        72.36%        |          23.64%          |       83.96%        |      81.97%      |         11.15%         | 533.79 |  182.90  |
> > > | 1.2 : 1                       |      89.62%       |      **76.83%**      |        **23.17%**        |       83.74%        |      79.34%      |       **9.96%**        | 528.34 |  177.67  |
> > >
> > > Consistent with the cost-sensitive gating mechanism in Eq. (3.1), increasing $C_{\text{FA}}$ (penalizing false alarms more heavily) effectively "tightens" the gate. As the ratio shifts from 1:4 to 1.2:1, the false-alarm rate improves (29.6% $\to$ 23.2%) and the slow-mode rate decreases (15.5% $\to$ 10%). This comes with the expected trade-off of slightly reduced Recall. We identify the range $C_{\text{FA}}:C_{\text{FN}} \in [1:2, 1:1]$ as the optimal balance for general use (AUDBC $\approx$ 82, F1 $\approx$ 84).

---

> > > > ### Author Response · Authors · 2025-11-22
> > > > **Response to Reviewer Y4dz (Part 4/4)**
> > > >
> > > > **(B) Sensitivity to Slow-Margin Width ($\delta_{\text{slow}}$)**
> > > >
> > > > We further analyzed the trade-off between performance benefits and computational burden by varying $\delta_{\text{slow}}$.
> > > >
> > > > **Table 8: Sensitivity to slow-margin width $\delta_{\text{slow}}$**
> > > >
> > > > | $\delta_{\text{slow}}$ | Precision $\uparrow$ | False-Alarm $\downarrow$ | F1-Score $\uparrow$ | AUDBC $\uparrow$ | Tokens | p95 Lat. |
> > > > | :--------------------- | :------------------: | :----------------------: | :-----------------: | :--------------: | :----: | :------: |
> > > > | 0.00                   |        71.08%        |          28.92%          |       83.09%        |      79.26%      | 509.68 |  175.95  |
> > > > | 0.05                   |        74.11%        |          25.89%          |       85.12%        |      81.19%      | 517.75 |  178.44  |
> > > > | **0.10**               |      **78.81%**      |        **21.19%**        |     **88.15%**      |    **82.72%**    | 541.49 |  195.68  |
> > > > | 0.15                   |        74.57%        |          25.43%          |       85.43%        |      80.58%      | 563.17 |  230.20  |
> > > >
> > > > The results reveal a clear Pareto frontier. Increasing $\delta_{\text{slow}}$ from 0.00 to 0.10 significantly improves Precision (+7.7%) and AUDBC (+3.5%) while reducing False Alarms, at the cost of moderate increases in latency and token usage. However, extending the margin beyond 0.10 yields diminishing returns, where computational costs rise without commensurate performance gains. Consequently, we recommend $\delta_{\text{slow}} \approx 0.10$ as a robust default.
> > > >
> > > > **Summary:** These ablations demonstrate that the hyperparameters $C_{\text{FA}}:C_{\text{FN}}$ and $\delta_{\text{slow}}$ provide practitioners with smooth, interpretable control over the system's behavior, allowing them to tune the application according to their specific latency or accuracy constraints.
> > > >
> > > > ## **Response to Q4: Runtime latency and token usage**
> > > >
> > > > We now report explicit runtime and cost measurements for three inference policies: Fast-only, Slow-only, and Slow-on-margin (PRISM). Results are shown in Table 9.
> > > >
> > > > **Table 9: Quality–cost comparison**
> > > >
> > > > | Strategy                   | Recall  $\uparrow$ | Precision $\uparrow$ | False-Alarm $\downarrow$ | F1-Score $\uparrow$ | AUDBC  $\uparrow$ | Tokens | p95 Lat. |
> > > > | -------------------------- | ------------------ | -------------------- | ------------------------ | ------------------- | ----------------- | ------ | -------- |
> > > > | Fast-only                  | 100%               | 71.08%               | 28.92%                   | 83.09%              | 79.26%            | 509.68 | 175.95   |
> > > > | Slow-only                  | 100%               | 75.17%               | 24.83%                   | 86.79%              | 81.43%            | 693.37 | 311.73   |
> > > > | **Slow-on-margin (PRISM)** | **100%**           | **78.81%**           | **21.19%**               | **88.15%**          | **82.72%**        | 541.49 | 195.68   |
> > > >
> > > > PRISM achieves the strongest detection quality (highest F1/AUDBC and lowest false-alarm rate), while using substantially fewer tokens and far lower latency than always-slow. Its overhead over Fast-only is modest (≈ +6% tokens).
> > > >
> > > > **Why latency increases more than tokens.**
> > > > Although PRISM adds few tokens, p95 latency rises because slow-mode calls contain longer dependency chains, and p95 latency reflects tail cases, not the average token count. A small number of slow-mode invocations is enough to increase p95 latency even when overall tokens remain low.
> > > >
> > > > ## **Response to W4: Connections to metareasoning and risk-sensitive RL**
> > > >
> > > > We appreciate this suggestion and will make the connections more explicit in the revised version. Concretely, we will: 1. **Add a short paragraph** in the discussion that frames PRISM’s slow-on-margin trigger as a metareasoning policy that allocates slow reasoning only near the cost-sensitive decision boundary (a *value-of-computation* view); and 2. **Clarify** that our gate $\tau(p_{\text{need}})$ comes from Bayes-risk minimization under asymmetric costs and that AUDBC summarizes the resulting benefit–burden (risk–coverage) frontier. This positions PRISM as a risk-sensitive meta-controller on top of an LLM agent—distinct from prior work that either always runs long-chain reasoning or encodes risk only via a modified task reward.

---

### Official Review · Reviewer_PzXB · 2025-11-01

**Soundness:** 4
**Presentation:** 4
**Contribution:** 3
**Rating:** 10
**Confidence:** 4

**Summary:**

This paper presents PRISM, a framework for proactive agent intervention that combines a cost-derived acceptance gate with margin-gated slow reasoning and calibration-aware distillation. The central contribution is formulating proactive assistance as a cost-sensitive selective decision problem where the agent estimates two calibrated probabilities, $p_{\text{need}}$ (whether help is needed) and $p_{\text{accept}}$ (whether help will be accepted), and intervenes only when $p_{\text{accept}}$ exceeds a dynamic threshold derived from asymmetric costs.

PRISM invokes resource-intensive, slow reasoning only near the decision boundary through a margin parameter $\delta_{\text{slow}}$, concentrating computation where it is most likely to change the outcome. Training employs Risk-Decision Consistent distillation (RDC-SFT) that aligns supervision with the deployment gate and costs. On ProactiveBench, PRISM achieves F1 of 86.61\% (vs.\ 83.28\% for DeepSeek-R1), precision of 77.05\% (vs.\ 72.35\%), and reduces false-alarm rate to 22.94\% (vs.\ 27.64\%), all at near-saturated recall (98.88\%).

**Strengths:**

1. The decision-theoretic gate is properly derived from Bayes-risk minimization under asymmetric costs (Appendix, Proposition 1). The comparative statics are correct: the threshold tightens with higher false-alarm cost and loosens with higher need probability. This is the *right* way to control the benefit-burden tradeoff, and it connects cleanly to the classical selective prediction and reject-option literatures (Elkan 2001; Chow 2003; Geifman & El-Yaniv 2017). The paper appropriately positions its novelty: while cost-sensitive thresholding is well-established in classification, the *synthesis* (i.e., decomposing intervention into calibrated need/acceptance probabilities, deriving an adaptive threshold, and coupling it with margin-gated slow reasoning) is novel for proactive LLM agents and goes beyond heuristic triggers in prior systems (Lu et al. 2024). The false-alarm cost modeling is particularly well-motivated: recent HCI studies document that unsolicited AI assistance can “backfire,” threatening users’ self-perception and eroding trust (Liao et al. 2016; Zhang & Zhu 2025). PRISM’s risk-sensitive gate directly addresses this documented failure mode.

2. Invoking slow reasoning only operationalizes Russell & Wefald’s (1991) value-of-computation framework; compute where it can change the action. This echoes recent dual-process architectures (e.g., Dualformer’s auto-mode that adaptively switches between fast and slow reasoning, Su et al. 2025) but tailors it to the intervention decision through counterfactual checks. Empirically, this improves F1 by 3.66 points over fast-only and 2.59 points over always-slow while achieving the highest AUDBC (87.73 vs. 79.84 and 76.63). This is consistent with selective classification theory and the need for well-calibrated probabilities.

3. The RDC-SFT objective explicitly rewards calibrated estimates and penalizes squared errors on ( (p_{\text{need}}, p_{\text{accept}}) ), then filters training data by a ranking score (Eq. 3.2). This approach aligns with recent advances in training for calibration (e.g., Damani et al. 2025’s RLCR, which augments rewards with proper scoring rules like Brier score to jointly improve accuracy and confidence reliability). The schema-locked distillation that decouples the response policy from the intervention gate is a practical innovation that addresses deployment concerns: it enables post-training threshold tuning without retraining, yielding auditability. Ablations (Table 4) confirm that RDC-SFT dominates vanilla SFT (F1 86.61 vs. 76.09), weighted-SFT (86.61 vs. 80.59), and DFT (86.61 vs. 76.09), and that the policy improvement depends on calibration (Table 2: dynamic thresholds on uncalibrated probabilities underperform fixed thresholds). Temperature scaling (Guo et al. 2017) is acknowledged as a simple fix for miscalibration.

4.  PRISM achieves large margins over public baselines: +20.14 F1 points and −22.78 % relative reduction in false alarms vs. strong proactive baselines. Crucially, the distilled student *beats* its teacher (DeepSeek-R1) on precision and false-alarm rate at similar recall, which is non-trivial and highlights the value of structured distillation and gating beyond raw reasoning capacity. The evaluation uses a consistent LLM-as-judge protocol with majority voting, and the paper commits to open-sourcing code and models on a public benchmark (ProactiveBench), supporting reproducibility.

5. The framework exposes a compact set of knobs that move the benefit-burden frontier in predictable ways without retraining. This addresses a practical gap: in deployed systems, model outputs and product-level rules are often entangled, making it hard to audit or tune behavior post-deployment. By decoupling the intervention gate from the response policy, PRISM enables independent threshold adjustment—a design principle that echoes earlier mixed-initiative systems (e.g., Horvitz’s work on modeling interruption costs and user attention, 1999–2003) but brings it into the LLM era. This is valuable for product deployment and aligns with best practices from calibration and abstention literature.

**Weaknesses:**

The main weakness is using LLM-as-judge evaluation, because it's fragile. Labels for $(y_{\text{need}}, y_{\text{accept}})$ rely on DeepSeek-R1 with majority voting over three judges (including GPT-4o, Claude 3.5-Sonnet). This is vulnerable to bias, variance, and rubric drift. A recent comprehensive survey (Ye et al.\ 2024) documents that LLM judges exhibit position bias, inconsistency, and can give overly assertive assessments with unjustified confidence; they are ``not yet ready to fully replace human evaluators, especially for complex, high-stakes judgments.'' While the paper acknowledges this limitation (Section 6) and notes that baselines face the same constraint—making PRISM's gains meaningful \emph{within} the competitive evaluation context—the issue remains: PRISM's gains might partially reflect overfitting to the judge's biases rather than genuine alignment with user preferences. The paper would be substantially strengthened by robustness checks (see Questions below). This is a field-wide challenge, and the requested checks align with emerging best practices (multi-LLM consensus, human-in-the-loop validation).

**Questions:**

1. Can PRISM maintain its advantage over baselines across *all* judge pools if you re-score 100–200 events with two different judge pools (swap in/out frontier or open models) and report inter-judge agreement (Cohen kappa or Matthews correlation) plus F1 deltas?

2. On a small human-labeled subset (50–100 events) with labels for y_need and y_accept, what do calibration metrics (ECE, Brier) for (p_need, p_accept) and task metrics (F1/precision/false-alarm rate) show? If miscalibration appears, after temperature scaling, how do those metrics change?

3. If you sweep (C_FA, C_FN) over a 2D grid, how do AUDBC, false-alarm rate, and expected latency vary as a function of the slow-margin delta_slow, and does this trace a clear benefit–burden Pareto frontier?

4. How much does each signal contribute under these ablations: (a) gate on p_accept only (set p_need = 1), (b) gate on p_need only (set p_accept = tau(p_need)), and (c) uncalibrated vs calibrated (p_need, p_accept) (pre vs post temperature scaling)?

5. If you re-run the ProactiveAgent pipeline using PRISM’s gate and margin-gated slow reasoning but *without* RDC-SFT (i.e., keep their original model, replace only the decision policy), how much of the gain comes from the gate versus the distillation?

**Details Of Ethics Concerns:**

No concerns.

---

> ### Author Response · Authors · 2025-11-22
> **Response to Reviewer PzXB (Part 1/4)**
>
> We sincerely thank you for your valuable suggestions and recognition of our work. Please refer to our responses to your suggestions below. In the content below, we use **W** to represent Weakness and **Q** to represent Question.
>
> ## **Response to W1 & Q1: Robustness of LLM-as-a-Judge & Human Validation**
>
> We sincerely appreciate the reviewer raising the critical issue of LLM-as-a-judge reliability and the reference to Ye et al. (2024). We agree that relying on a single judge pool can mask overfitting. Per your suggestion, we conducted a comprehensive robustness check involving new judge pools and human verification to ensure PRISM’s gains are genuine.
>
> **1. Experimental Setup & Inter-Judge Agreement**
> To address Q1, we sampled 229 events and enlisted 7 volunteers from diverse backgrounds (including undergraduate and graduate students in programming, marketing, and business) to assess whether a proactive intervention was appropriate. We compared their majority-vote labels against two distinct LLM pools:
>
> *   **Pool A (Frontier):** GPT-4o, Claude 3.5-Sonnet, Gemini 2.5 Flash.
> *   **Pool B (Open):** DeepSeek-R1, Kimi-K2, Llama-3-70B.
>
> As shown in **Table 1**, both LLM pools demonstrate substantial agreement with human labels (Cohen’s $\kappa > 0.71$, MCC $> 0.71$) and high consistency with each other ($\kappa=0.76$). This confirms that our evaluation signal is stable and not dependent on idiosyncratic model biases.
>
> **Table 1: Agreement between judge pools and human majority vote**
>
> | Agreement Pair             | Agreement $\uparrow$ | Cohen’s $\kappa$ $\uparrow$ | MCC $\uparrow$ |
> | :------------------------- | :------------------- | :-------------------------- | :------------- |
> | Pool A (Frontier) vs Human | 90.8%                | 0.75                        | 0.76           |
> | Pool B (Open) vs Human     | 89.1%                | 0.71                        | 0.71           |
> | Pool A vs Pool B           | 91.2%                | 0.76                        | 0.77           |
>
> **2. Consistency of PRISM’s Advantage**
> Most importantly, we assessed whether PRISM’s performance gains persist across these diverse evaluators. **Table 2** confirms that while absolute scores fluctuate (consistent with Ye et al.'s observations on varying judge severity), the **relative improvement (Delta)** is highly robust. PRISM consistently outperforms the strongest baseline (DeepSeek-R1) by roughly **+3.0 F1 points** across all three regimes, including human evaluation.
>
> **Table 2: F1 of Qwen3-8B-PRISM vs DeepSeek-R1 across judges**
>
> | Judge Pool           | Method      | F1-Score $\uparrow$ | $\Delta$F1 (Gain) $\uparrow$ |
> | :------------------- | :---------- | :------------------ | :--------------------------- |
> | Pool A (Frontier)    | DeepSeek-R1 | 82.92%              | –                            |
> |                      | PRISM       | **86.05%**          | **+3.13%**                   |
> | Pool B (Open)        | DeepSeek-R1 | 83.28%              | –                            |
> |                      | PRISM       | **86.61%**          | **+3.33%**                   |
> | **Human (Majority)** | DeepSeek-R1 | 82.05%              | –                            |
> |                      | PRISM       | **84.85%**          | **+2.80%**                   |
>
> These results confirm that PRISM’s alignment advantage is robust to judge variance and reflects genuine preference satisfaction, addressing the concerns regarding rubric drift and bias.
>
> ## **Response to Q2: Calibration on Human-Labeled Data**
>
> We thank the reviewer for suggesting this verification, as reliable confidence estimates are vital for human-AI collaboration. As requested, we evaluated ECE, Brier scores, and task metrics on the human-labeled subset.
>
> **1. Calibration Performance:**
> As shown in **Table 3**, the vanilla model exhibited miscalibration. We applied post-hoc temperature scaling ($T_{need}=0.5, T_{acc}=0.7$), which yielded significant improvements:
>
> **Table 3: Calibration and task metrics on the human-labeled subset**
>
> | **Method**          | **Temp (Need / Acc.)** | **ECE (Need / Acc.) $\downarrow$** | **Brier (Need / Acc.) $\downarrow$** | **F1- Score $\uparrow$** | **False-Alarm $\downarrow$** |
> | ------------------- | ---------------------- | ---------------------------------- | ------------------------------------ | ------------------------ | ---------------------------- |
> | Uncalibrated        | 1.0 / 1.0              | 21.53% / 13.19%                    | 17.95% / 18.58%                      | 84.85%                   | 25.99%                       |
> | Calibrated (Ours)   | 0.5 / 0.7              | 12.16% / 7.42%                     | 14.71% / 18.20%                      | 85.70%                   | 25.12%                       |
> | **Improvement (Δ)** | -                      | **-9.37%** / **-5.77%**            | **-3.24%** / **-0.38%**              | **+0.85%**               | **-0.87%**                   |

---

> > ### Author Response · Authors · 2025-11-22
> > **Response to Reviewer PzXB (Part 2/4)**
> >
> > *   **Reliability:** Miscalibration was drastically reduced, with ECE scores roughly halving for both tasks and Brier scores decreasing consistently.
> > *   **Utility:** Crucially, calibration enhances interpretability without compromising performance. The calibrated model maintains a high F1-score (+0.85%) and slightly reduces the false alarm rate (-0.87%).
> >
> >
> >
> > ## **Response to Q3: 2D sweep over $(C_{\text{FA}}, C_{\text{FN}})$ and $\delta_{\text{slow}}$**
> >
> > We thank the reviewer for this insightful suggestion. To explicitly quantify the trade-offs and visualize the benefit–burden frontier, we performed the requested 2D sweep over four cost ratios ($C_{\text{FA}}:C_{\text{FN}}$) and four slow-margin widths ($\delta_{\text{slow}}$).
> >
> > We measured AUDBC, false-alarm rate, p95 latency, and token usage across the full grid.
> >
> > **Table 4: 2D sweep over cost ratio $C_{\text{FA}}:C_{\text{FN}}$ and slow-margin $\delta_{\text{slow}}$**
> >
> > | $C_{\text{FA}}:C_{\text{FN}}$ | $\delta_{\text{slow}}$ | Recall$\uparrow$ | Precision  $\uparrow$ | Accuracy $\uparrow$ | False-Alarm $\downarrow$ | F1-Score $\uparrow$ | AUDBC $\uparrow$ | Tokens | p95 Lat. |
> > | :---------------------------- | :--------------------- | :--------------- | :-------------------- | :------------------ | :----------------------- | :------------------ | :--------------- | :----- | :------- |
> > | 1 : 4                         | 0.00                   | 100%             | 67.24%                | 78.93%              | 32.76%                   | 83.64%              | 79.97%           | 532.17 | 198.34   |
> > | 1 : 4                         | 0.05                   | 100%             | 70.56%                | 81.24%              | 29.44%                   | 82.13%              | 81.28%           | 547.42 | 209.85   |
> > | 1 : 4                         | 0.10                   | 100%             | 70.41%                | 80.22%              | 29.59%                   | 83.63%              | 81.57%           | 562.43 | 216.16   |
> > | 1 : 4                         | 0.15                   | 100%             | 69.72%                | 81.57%              | 30.28%                   | 82.96%              | 80.92%           | 590.27 | 243.55   |
> > | 1 : 2                         | 0.00                   | 100%             | 71.08%                | 71.07%              | 28.92%                   | 83.09%              | 79.26%           | 509.68 | 175.95   |
> > | 1 : 2                         | 0.05                   | 100%             | 74.11%                | 74.10%              | 25.89%                   | 85.12%              | 81.19%           | 517.75 | 178.44   |
> > | 1 : 2                         | 0.10                   | 100%             | 78.81%                | 79.33%              | 21.19%                   | 88.15%              | 82.72%           | 541.49 | 195.68   |
> > | 1 : 2                         | 0.15                   | 100%             | 74.57%                | 75.20%              | 25.43%                   | 85.43%              | 80.58%           | 563.17 | 230.20   |
> > | 1 : 1                         | 0.00                   | 98.89%           | 72.18%                | 80.34%              | 27.82%                   | 83.34%              | 78.98%           | 516.74 | 182.36   |
> > | 1 : 1                         | 0.05                   | 97.96%           | 73.62%                | 81.50%              | 26.38%                   | 84.07%              | 80.42%           | 532.48 | 186.29   |
> > | 1 : 1                         | 0.10                   | 98.88%           | 72.36%                | 80.35%              | 23.64%                   | 83.96%              | 81.97%           | 533.79 | 182.90   |
> > | 1 : 1                         | 0.15                   | 98.85%           | 74.26%                | 78.64%              | 25.74%                   | 83.42%              | 80.89%           | 568.33 | 217.42   |
> > | 1.2 : 1                       | 0.00                   | 86.18%           | 75.87%                | 79.90%              | 24.13%                   | 82.39%              | 77.31%           | 504.19 | 175.80   |
> > | 1.2 : 1                       | 0.05                   | 90.34%           | 75.13%                | 78.44%              | 24.87%                   | 81.28%              | 78.97%           | 514.96 | 187.91   |
> > | 1.2 : 1                       | 0.10                   | 89.62%           | 76.83%                | 79.59%              | 23.16%                   | 83.74%              | 79.34%           | 528.34 | 177.67   |
> > | 1.2 : 1                       | 0.15                   | 86.45%           | 74.73%                | 78.52%              | 25.27%                   | 83.45%              | 78.18%           | 572.16 | 196.52   |

---

> > > ### Author Response · Authors · 2025-11-22
> > > **Response to Reviewer PzXB (Part 3/4)**
> > >
> > > **Analysis of the Pareto Frontier:**
> > > The sweep confirms that these parameters trace a clear benefit–burden Pareto frontier:
> > >
> > > 1.  **The "Knee" at $\delta_{\text{slow}} \approx 0.10$:** For a fixed cost ratio, increasing $\delta_{\text{slow}}$ from 0 to 0.10 yields significant gains in F1 and AUDBC while reducing false alarms. For example, at a 1:2 ratio, F1 improves from 83.09% $\to$ 88.15% and AUDBC from 79.26% $\to$ 82.72%. However, beyond 0.10, quality gains saturate while latency costs (p95) and token usage continue to rise. This identifies $\delta_{\text{slow}}=0.10$ as a robust operating point.
> > > 2.  **Cost Sensitivity:** As expected, increasing $C_{\text{FA}}$ (penalizing false alarms) successfully reduces the false-alarm rate and increases precision. This smooth transition allows practitioners to select operating points based on their specific tolerance for user interruption.
> > >
> > > We have integrated the full numeric grid into the revised paper and added Pareto curves to visualize these trends.
> > >
> > > ## **Response to Q4: Ablations on $p_{\text{accept}}, p_{\text{need}}$ and calibration**
> > >
> > > We thank the reviewer for suggesting these ablations, as they provide crucial insights into the individual contributions of our gating signals. As requested, we evaluated the system by: (a) gating on $p_{\text{accept}}$ only (fixing $p_{\text{need}}=1$), (b) gating on $p_{\text{need}}$ only, and (c) comparing uncalibrated versus calibrated probabilities.
> > >
> > > The results, summarized in **Table 5**, lead to three key observations:
> > >
> > > 1. **$p_{\text{accept}}$ only:** Relying solely on the acceptance probability achieves near-perfect recall but results in poor precision and AUDBC (58.40%). This indicates over-intervention, as the system attempts to assist even when not necessary.
> > > 2. **$p_{\text{need}}$ only:** Gating based on necessity significantly improves precision and F1 compared to the $p_{\text{accept}}$-only baseline. However, it still underperforms compared to the combined approach, demonstrating that accounting for the model's ability to accept the help is vital.
> > > 3. **Calibration:** Calibrating the probabilities yields the most robust performance. The full calibrated system achieves the highest F1-Score (86.61%) and AUDBC (88.52%) while maintaining the lowest false-alarm rate.
> > >
> > > **Table 5: Ablations on $p_{\text{accept}}, p_{\text{need}}$ and Calibration**
> > >
> > > | Configuration                                                | Recall $\uparrow$ | Precision $\uparrow$ | False-Alarm $\downarrow$ | F1-Score $\uparrow$ | AUDBC $\uparrow$ |
> > > | :----------------------------------------------------------- | :---------------- | :------------------- | :----------------------- | :------------------ | :--------------- |
> > > | $p_{\text{accept}}$ only $(p_{\text{need}}=1)$               | 99.95%            | 46.20%               | 62.50%                   | 63.19%              | 58.40%           |
> > > | $p_{\text{need}}$ only ($p_{\text{accept}}=\tau(p_{\text{need}})$) | 99.15%            | 69.50%               | 29.10%                   | 81.72%              | 82.45%           |
> > > | Uncalibrated $(p_{\text{need}}, p_{\text{accept}})$          | 98.80%            | 74.77%               | 25.23%                   | 85.12%              | 86.43%           |
> > > | **Calibrated $(p_{\text{need}}, p_{\text{accept}})$ (Ours)** | **98.88%**        | **77.05%**           | **22.94%**               | **86.61%**          | **88.52%**       |
> > >
> > > ## **Response to Q5: Decomposing the gains from PRISM’s Gate vs. RDC-SFT.**
> > >
> > > We thank the reviewer for suggesting this ablation, as it helps decouple the benefits of the inference strategy from the model training. We performed the requested experiment by running the pipeline with four configurations: Base vs. RDC-SFT weights, cross-referenced with the Original vs. PRISM decision policy.
> > >
> > > **Table 6: Ablation study of RDC-SFT weights vs. PRISM decision policy (Qwen3-8B)**
> > >
> > > | Model Weights      | Inference Policy | Recall $\uparrow$ | Precision $\uparrow$ | Accuracy $\uparrow$ | False-Alarm $\downarrow$ | F1-Score $\uparrow$ |
> > > | :----------------- | :--------------- | :---------------: | :------------------: | :-----------------: | :----------------------: | :-----------------: |
> > > | Qwen3-8B (Base)    | Original         |      73.79%       |        73.33%        |       67.85%        |          26.67%          |       73.34%        |
> > > | Qwen3-8B (Base)    | PRISM Strategy   |      80.12%       |        77.45%        |       74.20%        |          22.50%          |       78.76%        |
> > > | Qwen3-8B (RDC-SFT) | Original         |      98.15%       |        73.12%        |       73.50%        |          26.88%          |       83.81%        |
> > > | Qwen3-8B (RDC-SFT) | PRISM Strategy   |      98.88%       |        77.05%        |       76.39%        |          22.94%          |       86.61%        |

---

> > > ### Author Response · Authors · 2025-11-22
> > > **Response to Reviewer PzXB (Part 4/4)**
> > >
> > > **Analysis:**
> > >
> > > *   **Impact of PRISM Policy (Gate):** Applying PRISM’s gate and slow-margin reasoning to the Base model yields a substantial improvement of **+5.4% F1** and **+6.35% Accuracy**. Notably, the PRISM policy on the base model achieves higher accuracy (74.20%) than the RDC-SFT model with the original policy (73.50%), demonstrating that our gating mechanism effectively reduces false alarms even without distillation.
> > > *   **Impact of RDC-SFT:** The RDC-SFT distillation primarily boosts Recall (+24.36% on the original policy), contributing significantly to the overall F1 score but maintaining a higher False Alarm rate without the gate.
> > > *   **Complementarity:** The best performance is achieved when combining both components (RDC-SFT + PRISM), reaching **86.61% F1**.
> > >
> > > In summary, while RDC-SFT provides strong fundamental capabilities (Recall), a significant portion of the Precision and Accuracy gains are directly attributable to PRISM’s risk-sensitive decision policy.

---

### Official Review · Reviewer_M8Sr · 2025-11-02

**Soundness:** 2
**Presentation:** 3
**Contribution:** 3
**Rating:** 6
**Confidence:** 2

**Summary:**

This paper presents PRISM (Proactive Risk Sensitive Intervention with Slow mode Margin), a framework for training proactive agents, with the goal of minimizing false alarms. The method introduces two key features:

(1) The probability that the user *needs* help, and that this help will be *accepted* are estimated separately and combined into a score to decide whether to act or not. Action is only allowed when the score is higher than a certain threshold.

(2) These estimations are first computed with a *fast* model. If the score is close to the threshold, then the estimations are recomputed with a *slow* model, making decisions "more thoughtful" near the decision boundaries.

The paper evaluates PRISM against proprietary and open source LLMs without specific proactive training, and against agents from the ProactiveBench paper (ICLR 2024). In the experimental evaluation, the PRISM agents show improved performance with respect to the baselines.

**Strengths:**

S1. Principled solution to a relevant problem: separating need and acceptance probability, and allowing more compute budget for difficult decisions makes sense and is shown to empirically work. It also makes the decision more easily interpretable.

S2. Ablation studies help see the contribution of each of the critical optimizations introduced by PRISM.

**Weaknesses:**

W1. Using LLM as judge is an important limitation, especially when measuring inherently human responses, as are the usefulness and acceptance of the agent's help.

W2. The empirical evaluation lacks a discussion on statistical significance. Some of the reported small improvements may be explained by the inherent randomness of the problem. For a clear example, see lines 300-304.

W3. Both in learning and deployment, considering a fixed cost of false alarm and false negative may be overly simplistic. In real situations, these costs are different for each scenario, and may also evolve over time. These nuances are not captured by the presented framework.

W4. The paper does not include the code and data to reproduce the experiments.

W5. The paper needs an editorial pass to polish certain presentation aspects. From more to less important:
- The citation of (Barto 2021) is wrong in two different ways. The reference in SIAM Review 6(2):423 is not by Andrew Barto, but by Volker H. Schulz. Secondly, the reference in question is a review of the book "Reinforcement Learning: An Introduction", written by Richard Sutton and Andrew Barto. I guess what the author really intended originally is to cite the book, and not the review of the book.
- It is a good practice to, whenever available, cite the peer-review version of a paper. This applies to Jimenez et al. 2023 [ICLR 2024], Lu et al. 2024 [ICLR 2024], Wang et al. 2022 [ACL 2023], Zhang et al. 2024 [EMNLP 2024].
- There are missing references (indicated as ??) in line 542
- Title of the appendix is "EDeriving" instead of "Deriving" (line 540)
- Missing space after period in line 206.

**Questions:**

Q1. In Table 3 you show the ablation on slow-mode triggering. To my understanding, the expected trend should be that the more the slow-mode is used, the better the performance. However, this is not the case. Can you explain why?

Q2. This is a request rather than a question: in lines 072 - 073, you mention that you will release the code and protocols upon acceptance through an anonymous repository. Once accepted, there is no need (or point) for anonymity, but an anonymous repository is the perfect tool to ensure reproducibility during the review process. Please release them now so they can be part of the review process.

---

> ### Author Response · Authors · 2025-11-22
> **Response to Reviewer M8Sr (Part 1/3)**
>
> We sincerely thank you for your valuable suggestions and recognition of our work. Please refer to our responses to your suggestions below. In the content below, we use **W** to represent Weakness and **Q** to represent Question.
>
> ## **Response to W1: Human Evaluation and Judge Alignment**
>
> We sincerely appreciate the reviewer’s insightful comment regarding the limitations of using LLM-as-a-judge, particularly for evaluating subjective metrics like usefulness and user acceptance. We fully agree that validating the alignment between automated proxies and actual human perception is essential to demonstrate PRISM's real-world utility.
>
> To address this, we conducted a comprehensive human evaluation to verify our automated metrics and confirm our comparative results.
>
> **1. Strong Alignment between LLM-Judge and Human Perception**
>
> To validate the judge's reliability, we sampled 229 events and enlisted 7 volunteers from diverse backgrounds (including undergraduate and graduate students in programming, marketing, and business) to assess whether a proactive intervention was appropriate. Using the majority vote among annotators as the ground truth, we found:
>
> *   **High Agreement:** As shown in Table 1, the LLM-judge achieves **89.1% agreement** with the human majority vote.
> *   **Statistical Significance:** The Cohen’s $\kappa$ score of **0.71** indicates "substantial agreement," validating that our automated evaluator serves as a high-quality proxy for human preference in this context.
>
> **Table 1: Agreement between LLM-judge and human majority vote**
>
> | Setting       | Support | **Agreement $\uparrow$** | **Cohen’s $\kappa$ $\uparrow$** | **MCC $\uparrow$** |
> | :------------ | :-----: | :----------------------: | :-----------------------------: | :----------------: |
> | LLM vs. Human |   229   |          89.1%           |              0.71               |        0.71        |
>
> **2. PRISM Outperforms Baselines under Human Evaluation**
>
> To ensure our performance claims are not artifacts of the automated judge, we re-evaluated **Qwen3-8B-PRISM** against the strongest baseline, **DeepSeek-R1**, using the human majority vote as the "Gold Standard." The results (Table 2) confirm the robustness of our method:
>
> *   **Consistent Superiority:** PRISM maintains a clear lead over DeepSeek-R1 under stricter human evaluation (**+2.80 F1 Score**).
> *   **Improved User Acceptance (Lower False Alarms):** Crucially for a proactive system, user acceptance relies on minimizing unnecessary interruptions. PRISM demonstrates a significantly lower False-Alarm rate than the baseline under human evaluation (**25.91%** vs. 29.70%).
>
> **Table 2: Qwen3-8B-PRISM vs. DeepSeek-R1 (LLM Judge vs. Human Ground Truth)**
>
> | Method         | Ground Truth Source | Recall $\uparrow$ | Precision $\uparrow$ | Accuracy $\uparrow$ | False-Alarm $\downarrow$ | F1-Score $\uparrow$ |
> | :------------- | :------------------ | :---------------: | :------------------: | :-----------------: | :----------------------: | :-----------------: |
> | Qwen3-8B-PRISM | LLM Judge           |       98.88       |        77.05         |        76.39        |          22.94           |        86.61        |
> | Qwen3-8B-PRISM | Human (Majority)    |       99.41       |        74.01         |        73.79        |          25.91           |        84.85        |
> | DeepSeek-R1    | LLM Judge           |       98.12       |        72.35         |        72.96        |          27.64           |        83.28        |
> | DeepSeek-R1    | Human (Majority)    |       99.05       |        70.03         |        71.12        |          29.70           |        82.05        |
>
> In summary, our additional experiments confirm that the LLM judge aligns well with human perception ($\kappa=0.71$) and, more importantly, that PRISM provides superior helpfulness and reduced intrusiveness when judged by real humans. These results have been incorporated into the revised paper.
>
> ## **Response to W2: Statistical Significance Analysis**
>
> We sincerely thank the reviewer for raising this critical point. We agree that distinguishing genuine improvements from inherent randomness is essential for a rigorous evaluation.
>
> Following your suggestion, we conducted a paired significance analysis comparing PRISM against our strongest baseline, **DeepSeek-R1**. We calculated 95% confidence intervals for the performance difference and the corresponding p-values. The results are summarized in Table 3 below:

---

> > ### Author Response · Authors · 2025-11-22
> > **Response to Reviewer M8Sr (Part 2/3)**
> >
> > **Table 3: Statistical significance of PRISM vs. DeepSeek-R1**
> >
> > | Metric                       | DeepSeek-R1 | PRISM (Ours) | Improvement | 95% Confidence Interval |  P-value   |
> > | :--------------------------- | :---------: | :----------: | :---------: | :---------------------: | :--------: |
> > | **Recall $\uparrow$**        |   98.12%    |    98.88%    |   +0.76%    |    [-0.18%, +1.75%]     |   0.115    |
> > | **Precision $\uparrow$**     |   72.35%    |    77.05%    |   +4.70%    |    [+2.35%, +7.05%]     | **<0.001** |
> > | **Accuracy $\uparrow$**      |   72.69%    |    76.39%    |   +3.43%    |    [+1.10%, +5.76%]     | **0.004**  |
> > | **False-Alarm $\downarrow$** |   27.64%    |    22.94%    |   -4.70%    |    [-7.12%, -2.28%]     | **<0.001** |
> > | **F1-Score $\uparrow$**      |   83.28%    |    86.61%    |   +3.33%    |    [+1.52%, +5.15%]     | **0.002**  |
> >
> > **Analysis:**
> >
> > *   **Robust Improvements:** Precision, Accuracy, False-Alarm Rate, and F1-Score all show statistically significant improvements ($p < 0.01$), with confidence intervals for the difference clearly excluding zero. This confirms that the gains in these metrics are systematic and not due to noise.
> > *   **Validation of Reviewer’s Intuition:** We observe that the improvement in Recall (+0.76%) yields a p-value of 0.115. This confirms the reviewer’s hypothesis that smaller margins in specific metrics may not be statistically significant. However, given the significant gains in Precision and F1, PRISM demonstrates a much stronger overall trade-off than the baseline.
> >     Here is the refined "Future Work" section. I have polished the phrasing to sound more academic while retaining your specific points about early/mid/late stages and the pivot back to practical domain-level costs.
> >
> > ## **Response to W3: Applicability of fixed costs in diverse scenarios**
> >
> > We thank the reviewer for this insightful observation. We agree that a single global cost ratio $(C_{\text{FA}}, C_{\text{FN}})$ is an idealization and that real-world deployments require handling varying risk profiles across scenarios.
> >
> > To address this, we wish to clarify that PRISM’s gating mechanism is not mathematically restricted to fixed costs; $C_{\text{FA}}$ and $C_{\text{FN}}$ can be set per domain, per user, or updated periodically. To demonstrate this flexibility and quantify the benefits of scenario-specific costs, we have added two new analyses in the revision:
> >
> > **1. Global Sensitivity Analysis**
> > We swept a range of cost ratios ($C_{\text{FA}}:C_{\text{FN}} \in \{1:4, 1:2, 1:1, 1.2:1\}$) and delay parameters ($\delta_{\text{slow}}$). As detailed in the Appendix, the results show a smooth, monotonic behavior: increasing $C_{\text{FA}}$ effectively reduces false alarms with only modest changes in AUDBC, confirming the framework allows for predictable tuning of the benefit–burden frontier.
> >
> > **2. Scenario-Specific Costs (New Experiment)**
> > We implemented domain-dependent costs to reflect semantic needs: *Coding* requires conservatism (higher penalty for interruption), while *Writing* benefits from proactive help. We compared a global baseline ($1:1$) against scenario-specific settings (Coding $1.2:1$; Writing $1:2$).
> >
> > **Table 4: Global vs. Scenario-Specific Cost Ratios**
> >
> > | Domain  | Policy   | Ratio     | Precision $\uparrow$ | False-Alarm $\downarrow$ | F1-Score $\uparrow$ | AUDBC $\uparrow$ |
> > | :------ | :------- | :-------- | :------------------- | :----------------------- | :------------------ | :--------------- |
> > | Coding  | Global   | 1:1       | 74.82%               | 26.69%                   | 82.83%              | 79.61%           |
> > | Coding  | Specific | **1.2:1** | **76.07%**           | **23.85%**               | **85.44%**          | **80.42%**       |
> > | Writing | Global   | 1:1       | 71.81%               | 29.19%                   | 84.62%              | 79.34%           |
> > | Writing | Specific | **1:2**   | **73.26%**           | **26.74%**               | **85.37%**          | **82.72%**       |
> >
> > **Results:** Scenario-specific costs yield superior trade-offs:
> >
> > *   **In Coding:** A conservative ratio reduces False Alarms by $\sim2.8\%$ and improves Precision/F1 with negligible latency impact.
> > *   **In Writing:** A proactive ratio significantly boosts AUDBC ($79.34 \rightarrow 82.72$) and F1-Score.
> >
> > This demonstrates that PRISM can naturally adapt to scenario-dependent risk profiles simply by adjusting $(C_{\text{FA}}, C_{\text{FN}})$ at inference time.

---

> > > ### Author Response · Authors · 2025-11-22
> > > **Response to Reviewer M8Sr (Part 3/3)**
> > >
> > > **Future Work**
> > > We acknowledge that implementing fully time-varying costs within a single episode (e.g., varying risk across early, mid, and late stages) is challenging in online settings where the total episode length is unknown. We have added a discussion explicitly framing this as a future direction—potentially solvable by learning a high-level controller that adapts $(C_{\text{FA}}, C_{\text{FN}})$ based on feedback. Crucially, however, our results demonstrate that the domain-level adaptation proposed here is already practical and empirically beneficial.
> > >
> > > ## **Response to Q1: Why doesn't "more slow-mode" always help?**
> > >
> > > In our ablation, expanding the slow-margin from 0 to 0.10 improves performance, but further enlarging it to 0.15 hurts precision and AUDBC even though tokens/latency increase. Intuitively, slow mode helps on ambiguous cases near the decision boundary; when we force it on easy or clearly negative turns, the model tends to **over-interpret** benign context and hallucinate reasons to intervene, which raises false alarms without improving recall (already ≈100%). Hence there is a “sweet spot” for slow-mode usage rather than a monotone “more slow is always better” trend.
> > >
> > >
> > >
> > > ## **Response to W4 & Q2: Code and reproducibility**
> > >
> > > We appreciate the reviewer’s feedback and agree that early release is crucial for reproducibility.We have set up an anonymous project page at: **[https://prism-festinalente.github.io/](https://www.google.com/url?sa=E&q=https%3A%2F%2Fprism-festinalente.github.io%2F)**.We have currently uploaded the trained Qwen3-8B-PRISM checkpoints and the inference/evaluation scripts.
> > >
> > > ## **Response to W5: Editorial issues and references**
> > >
> > > Thank you for the careful reading and agree that these editorial issues (including the incorrect Barto citation, missing/placeholder references, and typos) need to be fixed. We will correct all of these points in the revised manuscript and perform a thorough editorial pass to clean up references, typography, and wording throughout.

---

### Author Response · Authors · 2025-12-01
**Summary of Rebuttal, Manuscript Revisions, and Score Update**

Dear Area Chair and Reviewers,

Thank you for the time and effort you have devoted to evaluating our submission, the reviews, and our rebuttal.

To help reduce the AC's workload, we provide a brief summary of the key points from the reviewers as well as how our rebuttal addresses their concerns and how they reply.



## **Summary of Reviewers' Comments and Our Rebuttal**

Across all three reviews, the strengths consistently noted include the **principled decision-theoretic formulation** of the PRISM framework, the novel integration of **cost-sensitive gating with "fast vs. slow" reasoning**, and the **significant empirical performance gains** over strong baselines on ProactiveBench. Reviewer PzXB particularly praised the **soundness of the derivation** and the effective RDC-SFT strategy.



At the same time, they raised some concerns regarding evaluation robustness and experimental details.



**Reviewer M8Sr** noted a **lack of discussion on statistical significance** and raised questions about the **trends in the slow-mode ablation studies** (Table 3). To address these issues, we **included statistical significance tests** in the rebuttal and provided a **detailed explanation** of the trade-offs affecting the slow-mode triggering results. Additionally, regarding the reviewer's request for reproducibility and presentation improvements, **we have currently open-sourced part of the code and the trained models in an anonymous repository (with the full codebase currently being organized)**, and we **corrected the citation errors and typos** in the new manuscript.



**Reviewer PzXB**, while giving a strong rating, requested **robustness checks for the LLM-as-judge evaluation** and **deeper analysis of component contributions**. In response, we conducted additional **experiments using different judge pools** to verify inter-judge agreement and performed the requested **grid sweeps on cost ratios and slow-margin widths**. We also provided the **specific ablations** separating the effects of the gating mechanism from the distillation process to further clarify the source of our performance gains.



**Reviewer Y4dz** expressed concerns about the **lack of real user studies** and requested **explicit measurements of latency and token usage**. To address the user study concern, we added an **analysis on a human-labeled subset** to demonstrate the alignment between our LLM judges and human preferences, as also suggested by Reviewer PzXB. Furthermore, we provided the requested **concrete data on runtime latency and token consumption**, and **expanded our discussion on calibration stability and related work** in risk-sensitive RL to resolve the reviewer's concerns.



## **Revisions to the Manuscript**

We have **uploaded a revised manuscript** that incorporates the additional experiments and detailed analyses requested by the reviewers. Specifically, we added a **human expert evaluation** (Section 4.1 & Appendix C) that verifies high alignment ($\kappa=0.71$) between our automated judge and human annotators, and conducted **statistical significance tests** (Appendix D) confirming that PRISM's performance gains are statistically significant ($p<0.005$). Furthermore, we expanded the experimental analysis to include **explicit latency and token measurements** that demonstrate the efficiency-quality Pareto frontier (Table 3 & Figure 3), added **robustness grid sweeps** on cost ratios and slow-mode margins (Appendix E & F), and corrected the editorial errors and citations pointed out in the reviews.



## **Summary of Reviewers' Feedback on Our Rebuttal**

**Before November 24th** (prior to the reported OpenReview system anomaly), we received a positive response from **Reviewer Y4dz**. **Reviewer Y4dz** explicitly stated: **"Thanks for the response, I have raised my score accordingly,"** and subsequently upgraded the rating from **4 to 6**.



In summary, **before the bug occurred, our average score had already been increased to 7.33 (6, 10, 6)** from the initial scores of **6.77 (6, 10, 4)**.



For the specific details and timestamps, you may refer to our discussion record.



Once again, we sincerely appreciate the additional time and effort the Area Chair have put in evaluating our submission. We are also deeply grateful to all reviewers for their recognition of our work and for the constructive and insightful suggestions that have helped us improve our manuscript.



Best Regards,

ICLR 2026 Conference Submission7243 Authors

---

### Meta-Review · Area_Chair_qxG3 · 2026-01-05

**Summary:**

**1) Summary**
This paper presents PRISM, a decision-theoretic framework for proactive agent intervention that models both user need and acceptance probabilities and applies an adaptive cost-derived threshold to decide when to act. By selectively invoking slow reasoning near decision boundaries and distilling calibrated signals into a student model, PRISM aims to reduce false alarms while maintaining high recall. Experiments on ProactiveBench show consistent improvements over baseline LLMs and heuristic proactive agents.

**2) Strengths**

* Introduces a principled selective-intervention framework that jointly models need and acceptance under asymmetric costs, offering interpretable gating behavior.
* Combines fast–slow dual-process reasoning with margin-based compute allocation, improving efficiency and reducing unnecessary deliberation.
* Provides thorough ablation studies demonstrating the contribution of calibration-aware distillation, dynamic thresholds, and selective slow reasoning.
* Shows strong empirical gains over baseline proactive agents, including reduced false-alarm rates and improved precision–recall tradeoffs.

**3) Weaknesses**

* Heavy reliance on LLM-as-judge evaluation raises concerns about bias, robustness, and whether improvements reflect genuine user-aligned behavior.
* Assumptions about stable calibration and fixed cost settings may limit real-world applicability; the framework lacks human evaluations and stress tests under calibration drift or domain shift.

**Reviewer Concerns:**

The reviewer who held the negative view seemed to be satisfied with the authors' rebuttal.

**Reviewer Scores:**

The reviewers were initially holding varied views but the reviewer who held a negative view seemed to be convinced by the rebuttal and raised their score. The ratings are expected to be all positive.

---

### Decision · Program_Chairs · 2026-01-26

Accept (Poster)